# Development of microglia-targeting adeno-associated viral vectors as tools to study microglial behavior in vivo

Yukihiro Okada[1], Nobutake Hosoi [1], Yasunori Matsuzaki[1], Yuuki Fukai[1], Akito Hiraga[1], Junichi Nakai[2], Keisuke Nitta[1], Yoichiro Shinohara[1], Ayumu Konno [1,3] & Hirokazu Hirai [1,3 ✉]

Here we describe the microglia-targeting adeno-associated viral (AAV) vectors containing a 1.7-kb putative promoter region of microglia/macrophage-specific ionized calcium-binding adaptor molecule 1 (Iba1), along with repeated miRNA target sites for microRNA (miR)-9 and miR-129-2-3p. The 1.7-kb genomic sequence upstream of the start codon in exon 1 of the Iba1 (Aif1) gene, functions as microglia preferential promoter in the striatum and cerebellum. Furthermore, ectopic transgene expression in non-microglial cells is markedly suppressed upon adding two sets of 4-repeated miRNA target sites for miR-9 and miR-129-2-3p, which are expressed exclusively in non-microglial cells and sponged AAV-derived mRNAs. Our vectors transduced ramified microglia in healthy tissues and reactive microglia in lipopolysaccharide-treated mice and a mouse model of neurodegenerative disease. More-over, live fluorescent imaging allowed the monitoring of microglial motility and intracellular $Ca^{2+}$ mobilization. Thus, microglia-targeting AAV vectors are valuable for studying microglial pathophysiology and therapies, particularly in the striatum and cerebellum.

[1] Department of Neurophysiology and Neural Repair, Gunma University Graduate School of Medicine, Maebashi, Gunma 371-8511, Japan. [2] Division of Oral Physiology, Disease Management Dentistry, Tohoku University Graduate School of Dentistry, Sendai 980-8575, Japan. [3] Viral Vector Core, Gunma University, Initiative for Advanced Research, Maebashi, Gunma 371-8511, Japan. ✉email: hirai@gunma-u.ac.jp

Microglia, which are immune cells that reside in the central nervous system (CNS), are known to provide surveillance and scavenging functions. More recent studies, however, have revealed diverse functions of microglia in the CNS, including brain development[1,2], remodeling of neuronal circuits[1,3], and the progression of neurodegenerative diseases[4–6].

Recent technological advances have enabled the viral expression of miscellaneous functional molecules tagged with fluorescent proteins, such as G-CaMP (calcium sensor)[7] and ArcLightning (membrane voltage sensor)[8] for optical recording and diverse light-sensitive ion channels for optogenetics[9]. Under these circumstances, viral vectors that specifically transduce microglial cells in the brain in vivo are powerful tools for exploring the function and behavior of microglia in the native brain environment. In addition, microglia are chemoattracted to lesion sites in the brain;[10–12] thus, transduced microglia can be exploited to deliver a therapeutic gene product to the injured tissue.

To deliver a transgene to microglia in vivo, Jakobsson's group used lentiviral vectors with a housekeeping phosphoglycerate kinase (PGK) promoter and four repeated complementary microRNA-9-target (miR-9.T) sequences[13]. Since miR-9 was endogenously expressed in non-microglial cells but not in microglia, the transgene messenger RNA (mRNA) containing miR-9.T was sponged exclusively in non-microglial cells, leading to the selective transgene expression in microglia. Thereon, over 70% of transduced cells in the rat striatum were shown to be microglia[13]. However, switching the PGK promoter to a strong cytomegalovirus (CMV) promoter in the transgene cassette of lentiviral vectors causes prominent leakage of transgene expression in neurons and astrocytes[14]. In addition, relevant studies have shown the induction of miR-9 in microglia upon their activation[15–17], suggesting the possible suppression of transgene expression in reactive microglia.

Previous studies challenged the transduction of microglia using adeno-associated virus (AAV) serotype 2, 5 and mutant 6 capsid vectors comprising microglia-specific promoters, such as those of F4/80 and CD68[18,19]. They however resulted in only minor success with some "single microglia targeting," probably because of weak promoter activity and the low binding capacity of AAV capsids to microglia. Thus, for the efficient and selective transduction of microglia, a robust microglia-specific promoter and microglia-tropic AAV capsid are required. Among naturally isolated AAV capsid types, AAV1 and AAV9 in the cerebral cortex and AAV1 in the striatum caused high transgene expression in microglia. However, in addition to microglia, AAV1 in the striatum showed a preference also for astrocytes and neurons[20].

In this study, we developed microglia-targeting AAV2/9 vectors (AAV9 capsid and AAV2 viral genome) carrying a putative promoter region of the microglia/macrophage-specific ionized calcium-binding adapter molecule 1 (Iba1), miR-9.T, and additional target sequences complementarily bound by miR-129-2-3p, a different miRNA that was expressed selectively in non-microglial cell populations in the CNS[21–23].

## Results

To test microglial targeting, we produced five AAV constructs carrying the PGK promoter or putative Iba1 promoter with or without quadruplet miRNA target sequences for miR-9, miR-129-2-3p or miR-136-5p, as depicted (Fig. 1a–e). The microglia targeting profiles of these AAV constructs were assessed in three different brain regions: the cerebral cortex (frontal cortex), striatum, and cerebellum. The striatum was selected to compare our results with a previous report using lentiviral vectors with miR-9.T[13], while along with the striatum, the frontal cortex and

cerebellum were chosen as aberrant microglial function in these brain regions have been associated with various neurodegenerative disorders, such as Alzheimer's disease and cerebellar ataxias[24–26]. Generally, the transgene expression pattern of AAV depends on the injection dose and the subsequent incubation period. To screen for microglia-targeting properties, we waited a week after injection of AAV at a relatively high titer ($3.9 \times 10^{12}$ viral genome (vg) /mL; Fig. 1f). The injection volume (0.5 μL for the cerebral cortex, 1 μL for the striatum, and 10 μL for the cerebellum) was determined by several preliminary injection trials with reference to previous reports[13,27]. After choosing the most suitable AAV construct for microglia targeting, the incubation period was extended by lowering the injection dose to optimize microglial transduction specificity.

**Non-selective cellular transduction by AAV9.PGK.miR-9.T.** A previous study showed that the injection of lentiviral vectors expressing four repeated copies of miR-9.T by the PGK promoter in the adult rat striatum resulted in the predominant transduction of microglia[13]. The AAV2/9 vectors carrying the transgene cassette, which were essentially identical to the lentiviral vectors (AAV9.PGK.miR-9.T; Fig. 1a), were injected into the cerebral cortex, striatum, and cerebellum. One week after the injection, the brains were sliced and immunolabeled for GFP and Iba1. We found that the majority of GFP-expressing cells had morphologies distinct from ramified microglia and were not co-immunolabeled with Iba1 (Supplementary Fig. 1). Quantitative analysis showed that the ratios of GFP- and Iba1-positive cells to GFP-positive cells were ~10% or less in the cerebral cortex and striatum and 34% in the cerebellum (Table 1 and Supplementary Fig. 1c).

**Microglia targeting by AAVs carrying the Iba1 promoter in the striatum and cerebellum.** Previously, we successfully targeted specific cell populations using AAV vectors with cell type-specific promoters, such as the glial fibrillary acidic protein (GFAP) promoter (astrocytes)[28], neuron-specific enolase (NSE) promoter (neurons)[29], L7-6 promoter [cerebellar Purkinje cells (PCs)][30], and mouse glutamic acid decarboxylase (GAD) 65 (mGAD65) promoter (inhibitory neurons)[27]. Notably, we found that the genome sequence upstream of the first ATG of a cell type-specific gene often serves as a cell type-specific promoter[27,30]. Therefore, we decided to take advantage of this strategy. A previous study used a 1.9-kb Iba1 (Aif1) genomic region containing exon 1, intron 1, and a part of exon 2 to produce microglia-specific transgenic mice (Supplementary Fig. 2a)[31]. Thus, we tested whether the corresponding genomic region upstream of the first ATG in exon 1 of the Iba1 gene (Supplementary Fig. 2a, b) worked in a microglia-specific manner when incorporated into AAV vectors (hereafter, the 1678-bp mouse Iba1 genomic region is referred to as the Iba1 promoter). AAV2/9 vectors expressing GFP via the Iba1 promoter (AAV9.Iba1) were injected into the cerebral cortex, striatum, and cerebellum at the same dose as the AAV9.PGK.miR-9.T (Fig. 1b, f).

One week after the injection, efficient transduction of microglia (GFP and Iba1 double-positive cells) was observed in the striatum and cerebellum (Supplementary Fig. 3e–g and h–j), with some Iba1-negative non-microglial cells transduced (arrows in Supplementary Fig. 3f, g, i, j). In contrast, most GFP-expressing cells in the cerebral cortex were Iba1-negative pyramidal neuron-like cells (Supplementary Fig. 3b–d). Quantitative analysis showed that ~69% and ~86% of GFP-positive cells in the striatum and cerebellum, respectively, were Iba1-positive microglia, whereas Iba1-positive microglia retained only 2% of GFP-expressing cells in the cerebral cortex (Table 1 and Supplementary Fig. 3k). These

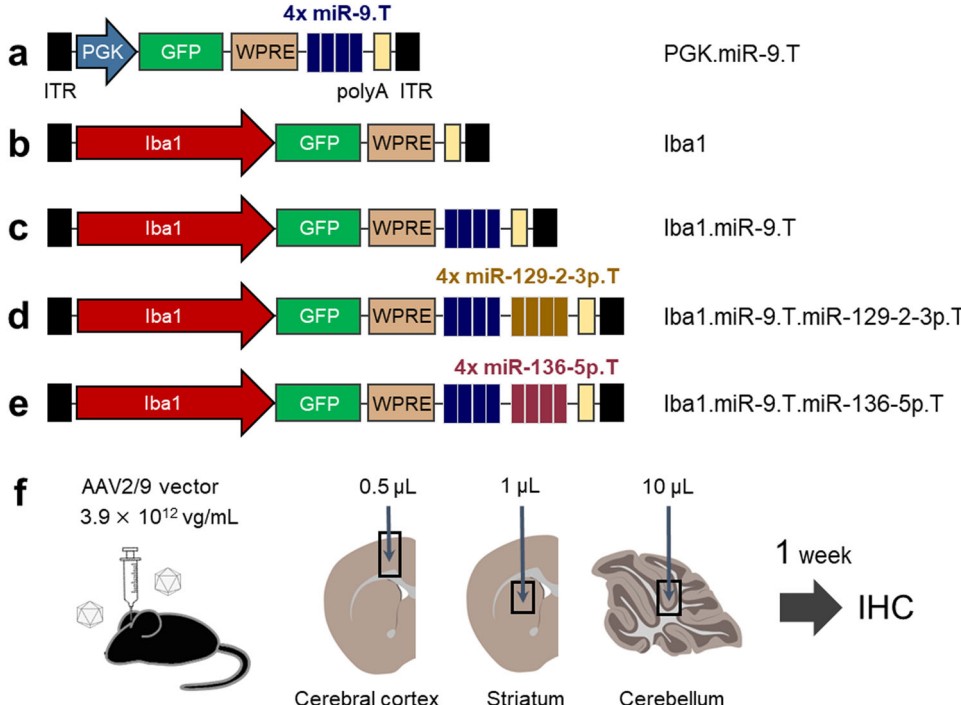

**Fig. 1 Schema of AAV constructs and experimental procedure. a–e** AAV vectors comprise the constitutive PGK promoter (**a**) or *Iba1* promoter (**b–e**) followed by GFP, WPRE, and polyA. In addition, one or two sets of four repeated microRNA (miR) target sequences (4 × miR-9.T and 4 × miR-129-2-3p.T or 4 × miR-136-5p.T) were inserted between the WPRE and polyA sequences (**a**, **c–e**). Each construct was labeled with the abbreviated promoter name (PGK or Iba1) plus incorporated miR target(s) as described. **f** Mice received different doses of each AAV vector (3.9 × 10$^{12}$ vg/mL) to the cerebral cortex (0.5 μL), striatum (1 μL), and cerebellum (10 μL). One week after the viral injection, the brains of the treated mice were examined by immunohistochemistry. GFP enhanced green fluorescent protein, Iba1 ionized calcium-binding adapter molecule 1, IHC immunohistochemistry, ITR inverted terminal repeat, PGK phosphoglycerate kinase 1, polyA simian virus 40 polyadenylation signal sequence, WPRE woodchuck hepatitis virus post-transcriptional regulatory element.

**Table 1 Microglia specificity of AAV vectors in different brain regions.**

| AAV construct | | Cerebral cortex | Striatum | Cerebellum | n |
|---|---|---|---|---|---|
| Promoter | miR target | | | | |
| PGK | miR-9.T | 12.0 ± 3.7 (319/2656) | 7.4 ± 3.4 (191/2576) | 34.4 ± 10.8 (901/2620) | 5 |
| Iba1 | – | 2.1 ± 1.3* (59/2791) | 68.8 ± 9.6** (1808/2628) | 85.7 ± 5.5 (2244/2618) | 5 |
| Iba1 | miR-9.T | 27.3 ± 2.4**† (753/2760) | 94.0 ± 2.0**† (2663/2833) | 100 ± 0*** (2726/2726) | 5 |
| Iba1 | miR-9.T miR-129-2-3p.T | 86.5 ± 5.6**†‡ (2750/3180) | 99.6 ± 0.5**† (3127/3140) | 100 ± 0*** (2712/2712) | 5 |

Data are presented as mean ± s.d. (%). Numbers in parentheses indicate the number of cells counted (GFP- and Iba1-positive cells/GFP-positive cells). Comparison of microglial specificity between AAV groups one week after the injection was assessed as follows: cerebral cortex, one-way ANOVA: $F_{(3,16)}$ = 540.425, $P < 0.001$; Bonferroni post hoc analysis: *$P = 0.003$, **$P \leq 0.001$ for vs. PGK.miR-9.T, †$P \leq 0.001$ for vs. Iba1, ‡$P \leq 0.001$ for vs. Iba1.miR-9.T; striatum, one-way ANOVA: $F_{(3,16)}$ = 330.221, $P < 0.001$; Bonferroni post hoc analysis: **$P \leq 0.001$ for vs. PGK.miR-9.T, †$P \leq 0.001$ for vs. Iba1; cerebellum, Kruskal–Wallis test, Kruskal–Wallis statistic = 18.348, $P < 0.001$; post hoc Dunn's analysis with Bonferroni adjustment for multiple comparisons: ***$P = 0.002$ for vs. PGK.miR-9.T. AAV adeno-associated virus, Iba1 ionized calcium-binding adapter 1, miR microRNA, n number of mice per group, PGK phosphoglycerate kinase 1.

results suggest that the *Iba1* promoter works preferentially in resident microglia in the striatum and cerebellum but not in the cerebral cortex.

**Non-microglia detargeting by addition of miR-9.T**. We added miR-9.T sequences to AAV9.Iba1 after the woodchuck hepatitis virus post-transcriptional regulation element (WPRE) (AAV9.Iba1.miR-9.T; Fig. 1c and Supplementary Fig. 4a). The miR target sequence was repeated four times as previous reports demonstrated that the levels of target suppression increased as the number of miR target sites increased[32–34]. The efficacy of each miR target site in the construct with four miR targets was two times larger than that of each site in the construct with two miR target sites, but was similar to that of the construct with six miR target sites[35]. We chose a four-time repeat in terms of efficacy for

target suppression and saving accommodation space for a transgene.

AAV efficiently transduced ramified microglia in all three brain regions (Supplementary Fig. 4b–j). However, in the cerebral cortex, numerous Iba1-negative non-microglial cells, characterized by large somata, were transduced around the epicenter of the viral injection site (Supplementary Fig. 4b, d), whereas selective microglial transduction was observed in a region away from the epicenter (Supplementary Fig. 4b, c). Quantitative analysis revealed that almost all GFP-positive cells in the striatum and cerebellum were Iba1-positive microglia, whereas the ratio of transduced microglia to total transduced cells was only ~27% in the cerebral cortex (Table 1 and Supplementary Fig. 4k). Therefore, addition of miR-9.T sequence significantly detargeted non-microglial cells in all three brain regions, although non-microglial detargeting in the cerebral cortex remained insufficient.

**Enhanced non-microglial detargeting by the addition of miR-129-2-3p target sequences to AAV9.Iba1.miR-9.T**. To further suppress transgene expression in non-microglial cells, we incorporated an additional quadruplex microRNA-129-2-3p target (miR-129-2-3p.T) or microRNA-136-5p target (miR-136-5p.T) sequences after quadruplex miR-9.T (AAV9.Iba1.miR-9.T.miR-129-2-3p.T or 136-5p.T) (Fig. 1d, e). Similar to miR-9, both miR-129-2-3p and miR-136-5p were shown to be enriched in neurons and diminished in microglia[22], suggesting that both miR-129-2-3p and miR-136-5p likely suppress the expression of a transgene mRNA containing miRNA target sites in non-microglial cells. We injected these AAVs similarly to three different brain areas and immunohistochemically analyzed the transgene expression profiles 1 week after the viral injection. Confocal microscopy analysis of the brain slices treated with AAV9.Iba1.miR-9.T.miR-129-2-3p.T revealed highly specific microglial transduction in all three brain regions (Fig. 2a–j). Notably, neurons in the cerebral cortex were largely detargeted (Fig. 2b–d). Quantitative analysis showed that ~87% of transduced cells in the cerebral cortex and almost all transduced cells in the striatum and cerebellum were microglia (Table 1). In contrast to miR-129-2-3p.T, the addition of miR-136-5p.T to AAV9.Iba1.miR-9.T failed to detarget the non-microglial cells in the cerebral cortex (Supplementary Fig. 5).

Next, we examined the transduction efficiency of microglia, which was assessed by dividing the percentage of GFP- and Iba1-double-positive cells (transduced microglia) by the total count of microglia immunoreactive for Iba1. AAV vectors at the dose tested transduced approximately 50 – 90% of microglia in all three brain regions (Supplementary Fig. 6). In the presence of miR-9.T, the transduction efficiency was significantly increased by switching from the PGK promoter to the *Iba1* promoter in the cerebral cortex and striatum. Addition of miR-129-2-3p.T to AAV9.Iba1.miR-9.T did not affect transduction efficiency in the three brain regions examined.

**Expression of GFP specifically in brain-resident microglia**. Direct parenchymal injection of AAV impairs the brain microvasculature, which may cause the entry of Iba1-positive blood-derived monocytes into the brain tissue. To verify whether the GFP-labeled cells contained these infiltrating monocytes, AAV9.Iba1.miR-9.T.miR-129-2-3p.T was injected into the striatum and cerebellum as depicted in Fig. 2. One week after the injection, striatal and cerebellar sections were triple immunolabeled for GFP, Iba1, and transmembrane protein 119 (TMEM119), a highly specific microglial marker that is not expressed by CNS macrophages, dendritic cells, infiltrating monocytes or other immune or neural cell types[36] (Supplementary Fig. 7). We examined GFP- and Iba1-double-positive cells ($n = 299$ cells in the five striatal sections from three mice and $n = 207$ cells in three cerebellar sections from two mice), all of which were also found to be positive for TMEM119, indicating little or no contamination of infiltrating monocytes in brain tissues injected with AAV9.Iba1.miR-9.T.miR-129-2-3p.T.

In addition, to confirm that the transduced cells were not astrocytes, striatal and cerebellar sections were immunostained for glial fibrillary acidic protein (GFAP), a marker for astrocytes (Supplementary Fig. 8). We carefully searched for GFP- and GFAP-double-positive cells (astrocytes) (five striatal sections from three mice and three cerebellar sections from two mice), but could not find such cells.

**Optimization of injection dose**. We examined the transduction profile three weeks after AAV injection (Fig. 3a). In the striatum, microglia-targeted transduction was preserved with faint GFP expression in non-microglial cells (Fig. 3b). Over 80% of GFP-positive cells were Iba1-positive microglia (80.6 ± 1.5%; 2061 cells in 2576 GFP-positive cells, $n = 9$ mice) (Fig. 3g). However, in the cerebellum, the specificity for microglia decreased to approximately 60% (56.1 ± 5.2%; 896 microglia in 1,637 GFP-positive cells, $n = 7$ mice) (Fig. 3c, g). Since the cerebellum received a 10-times higher dose of AAV than the striatum, we reduced the injection dose to one-fourth (from $3.9 \times 10^{10}$ vg/mouse to $1.0 \times 10^{10}$ vg/mouse), resulting in a significant increase in the microglia specificity to 83% (82.5 ± 4.9%, 890 cells in 1,067 GFP-positive cells, $n = 7$ mice) (Fig. 3d, g).

In the cerebral cortex, three-week incubation resulted in extensive transduction of non-microglial cells (4.0 ± 1.3%, 29 microglia in 718 GFP-positive cells, $n = 4$ mice; Fig. 3e, g), even though the cerebral cortex received the lowest dose of AAV. Further lowering the injection dose to one-fourth (from $1.95 \times 10^9$ vg/mouse to $0.5 \times 10^9$ vg/mouse) significantly increased microglial specificity, but still remained as low as 27.3 ± 3.5% (369 cells in 1316 GFP-positive cells, $n = 8$ mice; Fig. 3f, g). These results suggest that microglia in the striatum and cerebellum can be stably transduced with high specificity by choosing the optimal injection dose; however, it is still difficult to suppress transgene expression in non-microglial cells in the cerebral cortex.

**Transduction of reactive microglia in LPS-treated mice**. Previous studies have reported the induction of miR-9 production following lipopolysaccharide (LPS) treatment in microglia[17] and monocytes[15]. If this is the case, the AAV9.Iba1.miR-9.T.miR-129-2-3p.T may not be useful in LPS-activated reactive microglia. To validate this hypothesis, we injected AAV9.Iba1.miR-9.T.miR-129-2-3p.T at initial high dose (cerebral cortex, $1.95 \times 10^9$ vg/mouse; striatum, $3.9 \times 10^9$ vg/mouse; cerebellum, $3.9 \times 10^{10}$ vg/mouse) together with LPS (0.2 µg/µL) in adult mice, followed by intraperitoneal injection of LPS (1 µg/g body weight) every day for one week (Fig. 4a). The mice were killed for immunohistochemistry. We observed robust GFP expression in numerous microglia in all three brain regions (Fig. 4b–m). Transduced microglia in LPS-treated mice were amoeboid in shape, featuring shorter and thicker processes. These results indicate the effective transduction of reactive microglia, as well as ramified microglia, by AAV9.Iba1.miR-9.T.miR-129-2-3p.T.

**Transduction of reactive microglia in SCA1-Tg mice**. Microglia recognize the specific structure of LPS via TLR4 (Toll-like receptor 4), leading to the release of pro-inflammatory cytokines, while microglia in neurodegenerative tissues are likely stimulated via scavenger receptors, which induce phagocytosis of apoptotic cellular debris and the release of anti-inflammatory cytokines[37]. Thus, reactive microglia in neurodegenerative tissues may produce miRNAs that are distinct from those in LPS-exposed microglia. We then examined whether the AAV9.Iba1.miR-9.T.miR-129-2-3p.T can be used for the transduction of microglia in neurodegenerative tissues. As a neurodegenerative model, we chose spinocerebellar ataxia type 1 (SCA1) transgenic (SCA1-Tg) mice that express abnormally expanded ATXN1, specifically in cerebellar PCs under the control of the PC-specific L7 promoter (also known as the B05 line)[38,39]. Twenty-four-week-old symptomatic SCA1-Tg mice and their wild-type littermates received cerebellar injections of a lower dose of AAV9.Iba1.miR-9.T.miR-129-2-3p.T ($1.0 \times 10^{10}$ vg/mouse; Fig. 5a).

Three weeks after the viral injection, cerebellar sections were immunolabeled for GFP and Iba1. Confocal microscopy revealed efficient and specific transduction of microglia in sections from both SCA1-Tg mice and their wild-type littermates (Fig. 5b–g).

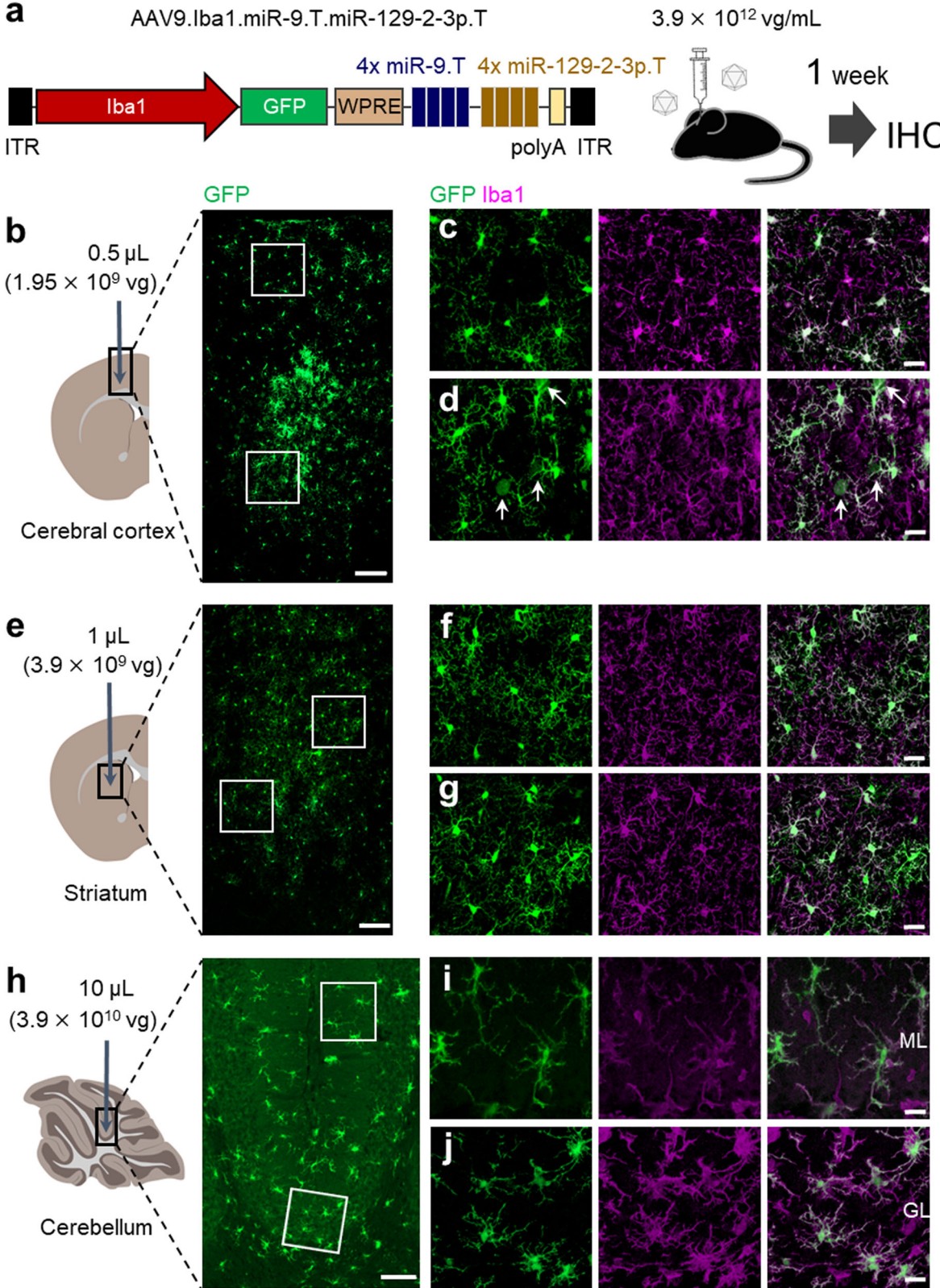

**Fig. 2 Highly efficient and specific microglial targeting by AAV9.Iba1.miR-9.T.miR-129-2-3p.T. a** Schema of the AAV construct comprising *Iba1* promoter, GFP, and two different quadruplex sequences complementary to miR-9 and miR-129-2-3p. Mice received different doses of AAV ($3.9 \times 10^{12}$ vg/mL) to the cerebral cortex (0.5 μL), striatum (1 μL) and cerebellum (10 μL). One week after the viral injection, the mouse brains were examined by immunohistochemistry. (**b-d**)Immunohistochemistry of the cerebral cortex. Square regions in **b** were enlarged. Note the strong and efficient GFP expression in microglia with weak and rare transduction of Iba1-negative non-microglial cells (arrows). **e–j** Highly specific transduction of microglia in the striatum (**e–g**) and cerebellum (**h–j**). Square regions in (**e**, **h**) were enlarged. Scale bars; 100 μm (**b**, **e**, **h**) and 20 μm (**c**, **d**, **f**, **g**, **i**, **j**). GL granule cell layer, ML molecular layer.

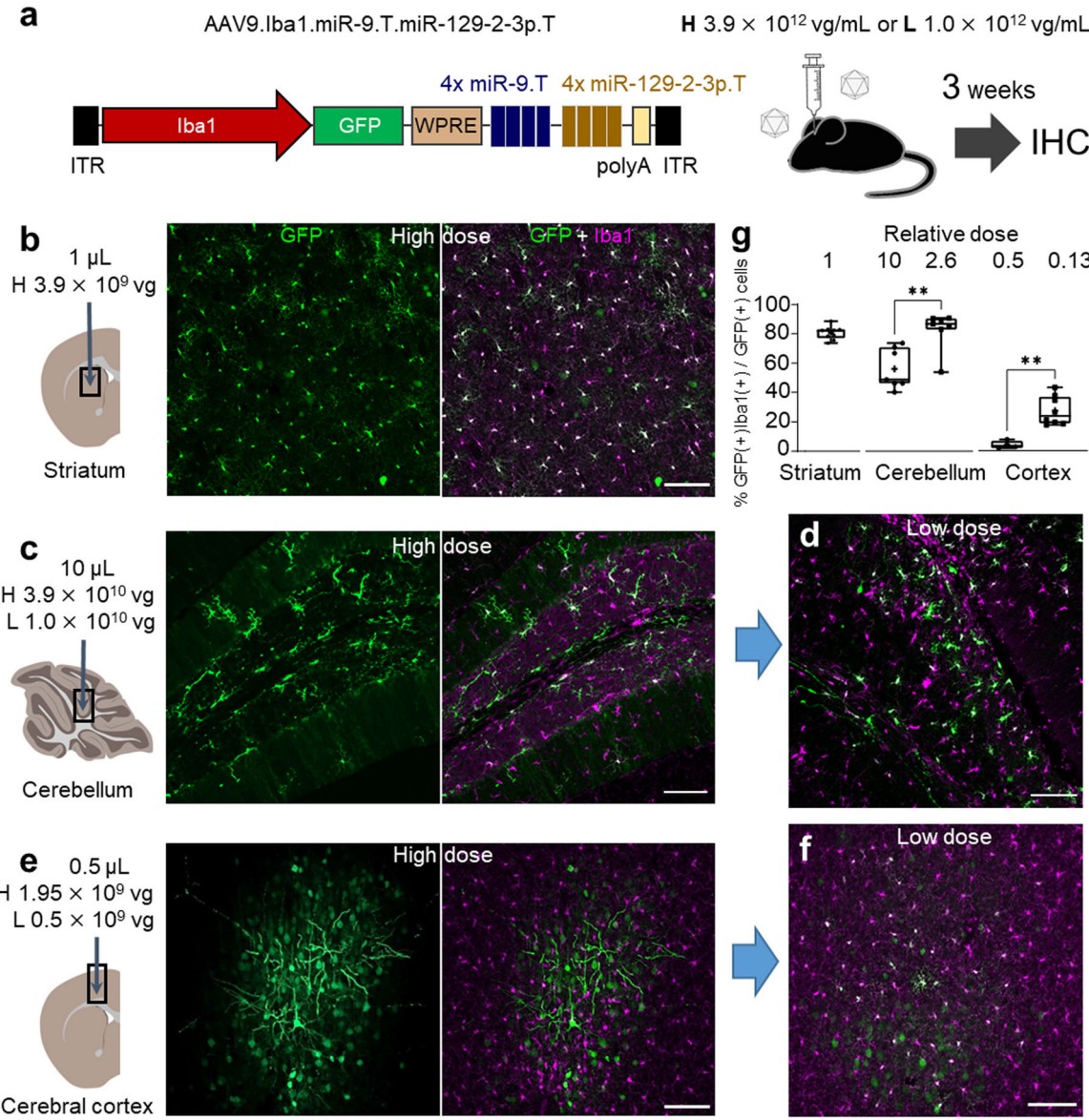

**Fig. 3 Maintenance of microglia-specific transduction by AAV injection at optimal doses. a** AAV construct (AAV9.Iba1.miR-9.T.miR-129-2-3p.T) and schematic of the virus injection. Mice that received an injection of AAV in three brain regions at the same dose as Fig. 2 (High dose; H) or about one quarter (Low dose; L) were examined 3 weeks post-injection by immunohistochemistry. **b**–**f** Immunohistochemistry of the striatum (**b**), cerebellum (**c, d**), and cerebral cortex (**e, f**). Injected doses were described above the schemas. Note the strong and efficient GFP expression in microglia with weak transduction of Iba1-negative non-microglial cells in the striatum and cerebellum (**b, c**). In contrast, the cerebral cortex showed GFP expression mostly in non-microglial cells including those displaying pyramidal neuron-like morphology (**e**). Lowering the viral dose to a quarter suppressed GFP expression in non-microglial cells in the cerebellum and cerebral cortex (**d, f**). Scale bars; 100 μm. **g** Summary graph showing the cell specificity for microglia in the three brain regions. Numbers above the graph represent relative doses of injected AAV when the dose injected to the striatum is taken as 1. Box-and-whisker plots display the median (centerline), 25th to 75th percentile (box), and minimum to maximum values (whiskers) (striatum, $n = 9$ mice per group; cerebellum, $n = 7$ mice per group, two-tailed unpaired $t$-test, $t_{(12)} = 3.711$, **$P \leq 0.01$ for high dose vs. low dose; cerebral cortex, $n = 4$ mice per group, two-tailed unpaired $t$-test, $t_{(6)} = 4.513$, **$P \leq 0.01$ for high dose vs. low dose).

Notably, a much higher population of microglia was transduced in the SCA1-Tg mouse cerebellum than in wild-type littermates, at least partly due to the significantly increased density of microglia compared with age-matched wild-type mice[40]. These results extend the AAV9.Iba1.miR-9.T.miR-129-2-3p.T for specifically targeting microglia in neurodegenerative tissues.

**Monitoring the process motility and Ca²⁺ dynamics in cerebellar microglia**. To examine the dynamic morphological

changes in microglia, confocal live GFP imaging was performed in the granule cell layer of acute cerebellar slices seven–12 days after AAV injection (AAV9-Iba1-GFP-miR-9.T-miR-129-2-3p.T; $3.9 \times 10^{10}$ vg/mouse) to the cerebellum. Most GFP-labeled microglial cells showed basal morphological motility, although the degree of motility varied between the cells (Fig. 6a; Supplementary Videos 1–3). In some cells, the bath application of ATP (100 μM) induced an increase in motility with some delay (Fig. 6b and Supplementary Videos 4 and 5). These results are in line with the morphological dynamics of microglia[41,42].

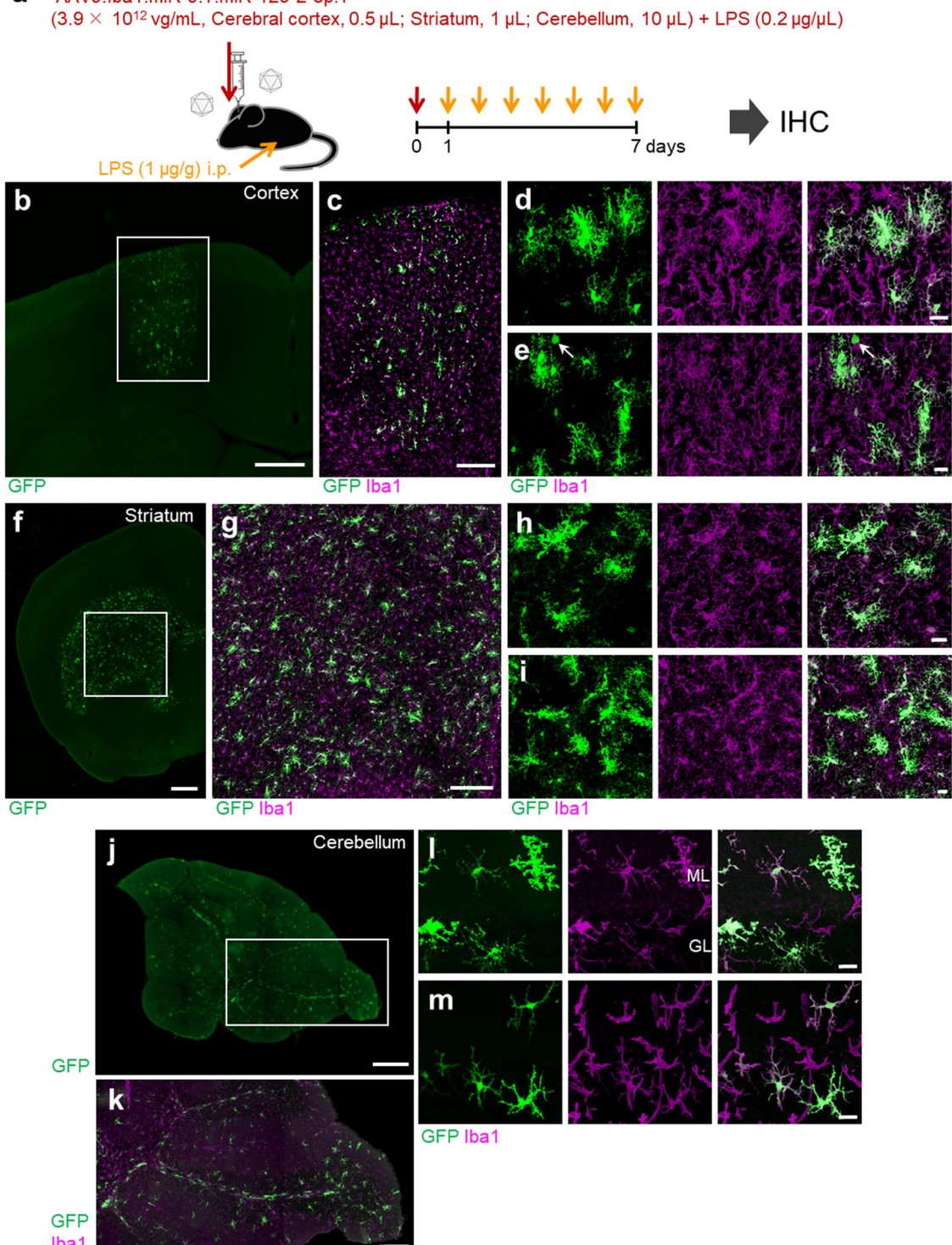

**Fig. 4 Robust transduction of microglia in LPS-treated mice. a** Diagram depicting the experimental protocol. A solution containing AAV9.Iba1.miR-9.T.miR-129-2-3p.T and LPS (0.2 μg/μL) was injected into the three brain regions at Day 0, followed by supplemental intraperitoneal injection of LPS (1 μg/g body weight) daily up to Day 7. Mice were killed 6 h after the LPS injection on Day 7. Brain slices were immunolabeled for GFP and Iba1. **b–m** Low, middle, and high magnification of the cerebral cortex (**b–e**), striatum (**f–i**), and cerebellum (**j–m**). Square regions in (**b**), (**f**), and (**j**) were enlarged to (**c**), (**g**), and (**k**), respectively. Scale bars: 500 μm (**b**, **f**, **j**), 200 μm (**c**, **g**, **k**), and 20 μm (**d**, **e**, **h**, **i**, **l**, **m**). GL granule cell layer, LPS lipopolysaccharide, ML molecular layer.

Next, we examined whether our microglia-selective gene expression method could be used to measure $Ca^{2+}$ signal from a genetically encoded fluorescent calcium indicator, G-CaMP7.09[43]. At least one week after AAV injection (AAV9-Iba1-G-CaMP7.09-miR-9.T-miR-129-2-3p.T; $3.9 \times 10^{10}$ vg/mouse) to the cerebellum,

live confocal $Ca^{2+}$ imaging was performed in the granule cell layer of acute cerebellar slices. Some microglia showed spontaneous $Ca^{2+}$ activity (Supplemental Videos 6–8; before ATP application, and Fig. 7b; ROI 3 before ATP application). Bath application of ATP (100 μM) reliably elicited $Ca^{2+}$ increase not only in larger

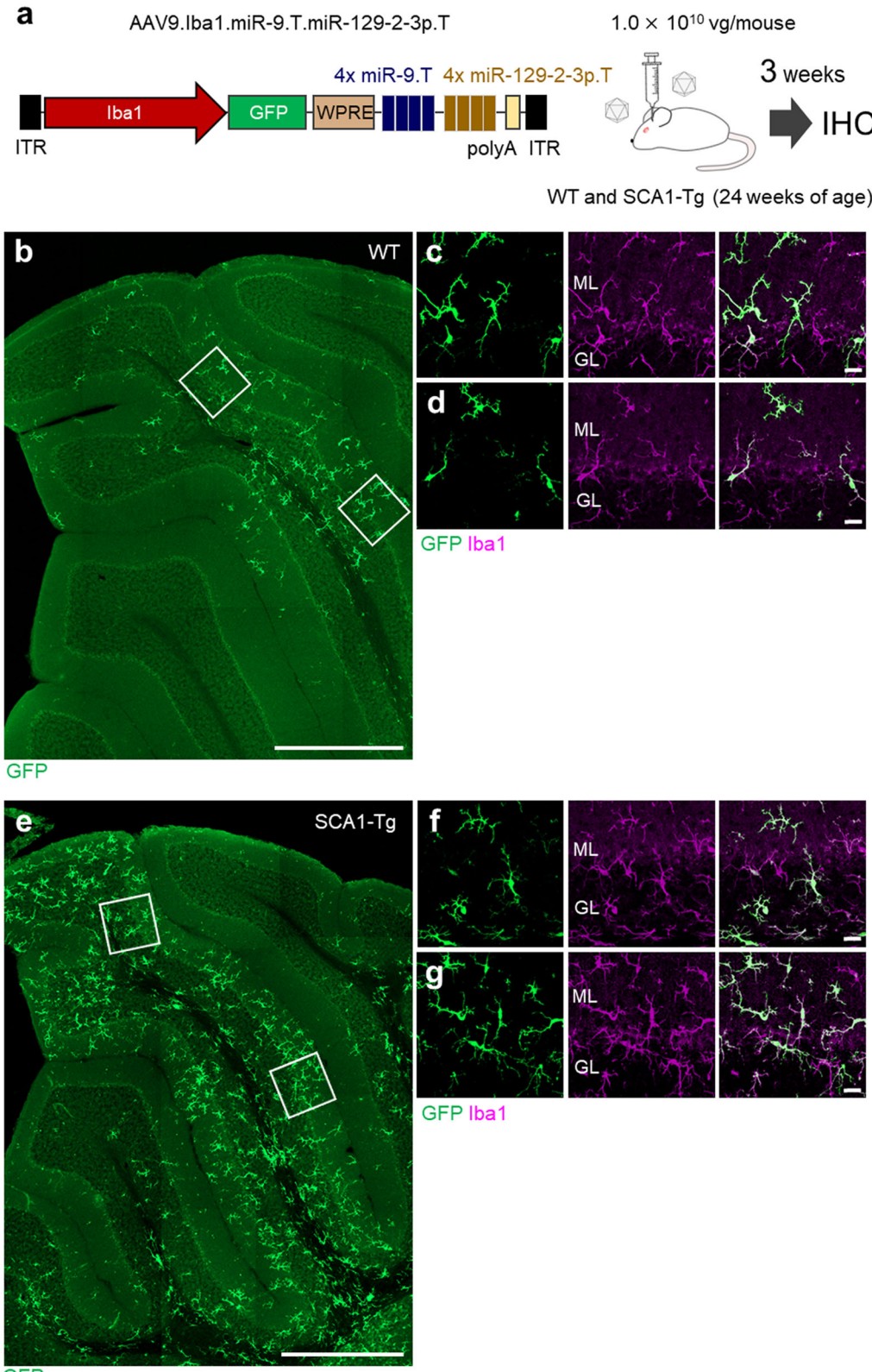

**Fig. 5 Robust transduction of microglia in symptomatic SCA1-Tg mouse cerebellum. a** AAV construct (AAV9.Iba1.miR-9.T.miR-129-2-3p.T) and schematic of the virus injection. AAV (dose: $1.0 \times 10^{10}$ vg/mouse) was injected into the wild-type and SCA1-Tg mouse cerebellum at 24 weeks of age. Three weeks after the viral injection, cerebellar sections were analyzed by immunohistochemistry. **b–d** GFP immunofluorescent image of the wild-type (WT) mouse cerebellum. **e–g** GFP immunofluorescent image of a sagittal section of the SCA1-Tg (B05) mouse cerebellum. Square regions in **b** and **e** were enlarged to **c**, **d** and **f**, **g**, respectively. Scale bars; 500 μm (**b**, **e**) and 20 μm (**c**, **d**, **f**, **g**). GL granule cell layer, ML molecular layer.

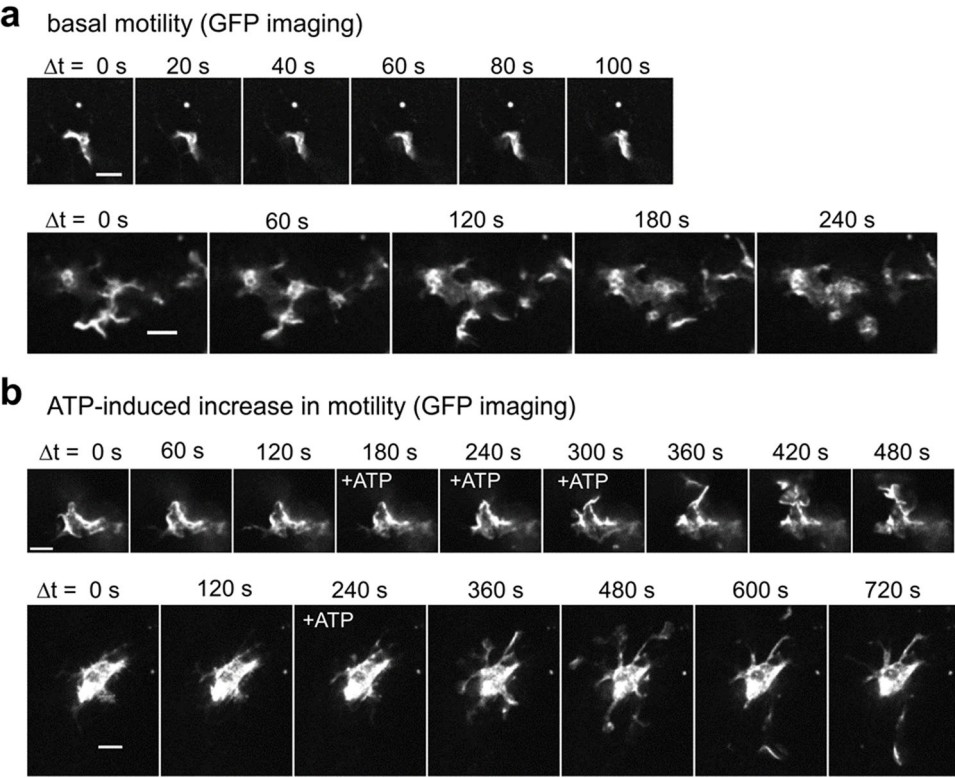

**Fig. 6 Motility analysis of cerebellar microglia overexpressing GFP with our microglia-selective gene expression method. a, b** Time-lapse GFP fluorescent images (time indicated above the images) showing basal movements of microglial processes in **a** and ATP-induced motility upregulation of microglia in **b** (frames, when 100 μM ATP was bath-applied, are indicated as '+ATP'; 3 min ATP application in the upper panel and 1 min in the lower panel of **b**). The Upper and lower panels in **a** and **b** correspond to examples from different cells. Scale bars: 10 μm.

cellular compartments (Fig. 7a, presumably microglial somas or larger microglial processes; Supplementary Videos 7 and 8) but also in smaller compartments (Fig. 7b, d, presumably microglial fine processes; Supplemental Video 7) of G-CaMP-positive cells. The mean peak amplitude of ATP-induced $Ca^{2+}$ signal changes ($\Delta F/F_{basal}$, see methods) was $3.36 \pm 0.19$ (Fig. 7c, $n = 91$ cellular compartments from five cerebellar slices of two mice). These results match typical microglial $Ca^{2+}$ dynamics because microglia express purinergic receptors and exhibit ATP-induced $Ca^{2+}$ responses on the order of seconds[35,44].

Taken together, these results suggest that the functional $Ca^{2+}$ indicator G-CaMP proteins are successfully expressed in cerebellar microglia using this method. Thus, we conclude that our AAV-mediated microglia-specific gene expression method is also applicable to the expression of a gene other than GFP, and that our method can be a powerful tool to examine live morphological dynamics in microglia.

**Overexpression of miR-9.T and miR-129-2-3p.T throughout the brain using AAV-PHP.B.** Cellular microRNAs, including miR-9 and miR-129-2-3p, play critical roles in cellular functions such as cell survival, dendritic branching, and synaptic plasticity[45–49]. Entrapment of endogenous miRNAs by over-expressed miR.T sequences in neurons and/or other cell types may impair the inherent roles of miRNAs and consequently disrupt cellular functions. To test this possibility, mRNA containing miR-9.T and miR-129-2-3p.T was overexpressed throughout the brain by the systemic application of the blood-brain-barrier-penetrating AAV-PHP.B expressing GFP-miR-9.T-miR-129-2-3p.T, using the powerful cytomegalovirus and chicken β-actin hybrid (CBh) promoter[50] (AAV-PHP.B-CBh-GFP-miR.T) or AAV-PHP.B expressing GFP alone using the same CBh promoter (AAV-

PHP.B-CBh-GFP; Supplementary Fig. 9a). Since the cerebellum expresses higher levels of miR-9 and miR-129-2-3p than other brain regions such as the cerebral cortex, brainstem, and spinal cord[51], we tested the cerebellum-associated motor coordination ability of the treated mice using rotarod and beam-walking tests three weeks after viral injection. The results showed no statistically significant difference between the two groups in either behavioral test ($t$-test, $n = 5$ mice for each group; Supplementary Fig. 9b, c).

After the behavioral tests, mice were killed for histological analysis. Control mice showed robust GFP expression in both the liver and the brain ($n = 5$ mice, middle panels in Supplementary Fig. 9d). In contrast, mice expressing GFP-miR-9.T-miR-129-2-3p.T showed almost no GFP expression in the brain, despite strong GFP expression in the liver ($n = 6$ mice; lower panels in Supplementary Fig. 9d). Sagittal sections of the brain confirmed extensive suppression of GFP expression throughout the brain by the co-expression of miR-9.T-miR-129-2-3p.T (top right panel in Supplementary Fig. 9e). Enlarged images show faint GFP expression in a small number of cells, including vascular endothelial cells and cerebellar PCs (middle and bottom right panels in Supplementary Fig. 9e).

**No significant changes in endogenous miRNA levels were observed upon overexpression of miR-9.T and miR-129-2-3p.T.** Endogenous miRNAs bound to miR target sequences may be degraded along with viral mRNAs. To examine this, we compared the levels of endogenous miR-9, miR-129-2-3p, and miR-129-5p (as a control) in mice expressing GFP-miR-9.T-miR-129-2-3p.T, and those expressing GFP alone in three brain tissues (cerebral cortex, striatum, and cerebellum). The results showed no statistically significant differences in miR-9, miR-129-2-3p,

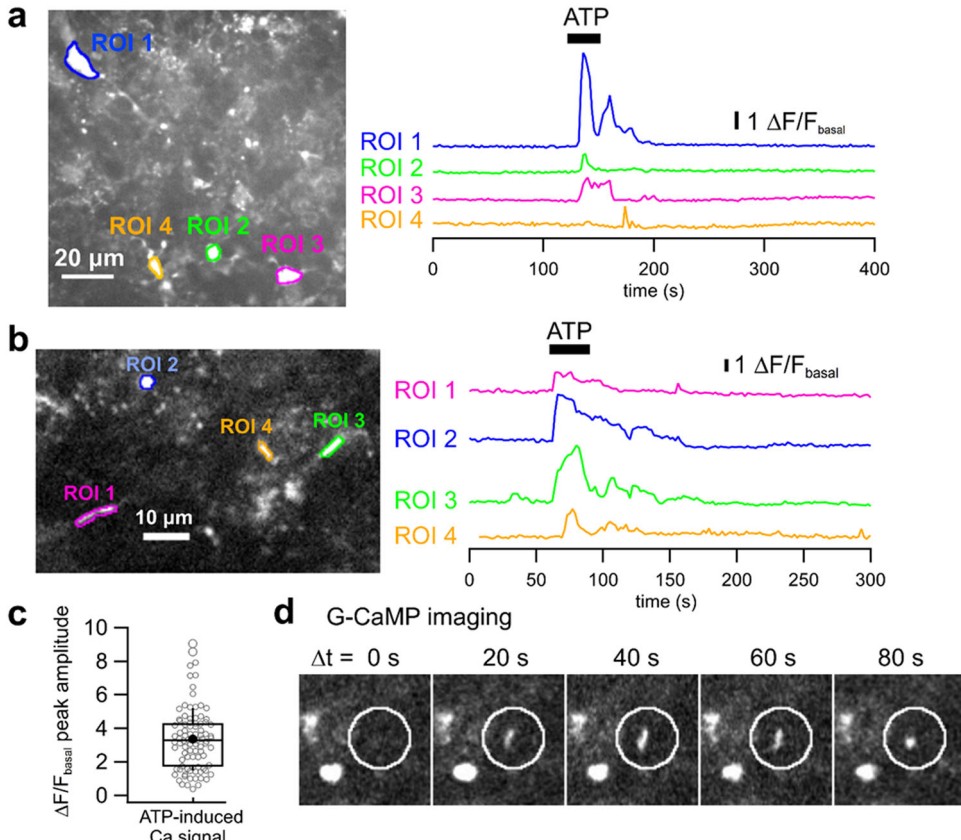

**Fig. 7 Application of our microglia-selective gene expression method in expressing the genetically encoded Ca$^{2+}$ indicator G-CaMP7.09 and measuring Ca$^{2+}$ signals in cerebellar microglia. a, b** Left panels show the averaged confocal images of virally expressed G-CaMP signals in the granule cell layer of acute cerebellar slices. Regions of interest (ROIs 1–4) were set on larger compartments (possibly microglial somata or their larger processes) in *a* and on smaller compartments (possibly fine microglial processes) of the microglia in **b**. The right panels show traces of Ca$^{2+}$ signal changes estimated from the fluorescence of the ROIs illustrated in the left panels. Bath application of 100 μM ATP (indicated in black bars) evoked Ca$^{2+}$ transients robustly in both the larger and the smaller compartments of the transduced microglia. **c** A box and whisker plot showing quantified Ca$^{2+}$ signals induced by bath-applied ATP (ΔF/F$_{basal}$, see methods) in the microglial cellular compartments. Open circles indicate individual data points. The horizontal line and the box represent the median value and the interquartile range, respectively. The error bars indicate one standard deviation above and below the mean value (filled circle). **d** Time-lapse fluorescent images showing the movement of a microglial process expressing G-CaMP7.09 (indicated by a white circle). Note that the other G-CaMP-positive microglial compartments outside the white circle were static.

and miR-129-5p levels between the two groups in all three brain regions (*t*-test, *n* = 5 mice per group, Supplementary Fig. 9f).

**No significant impairment of the electrophysiological functions of cerebellar Purkinje cells upon overexpression of miR-9.T and miR-129-2-3p.T.** To verify the influence of miR.T overexpression on neural function, we examined the electrophysiological properties of the cerebellum in AAV-PHP.B-treated mice expressing GFP alone or in combination with miR.T (miR-9.T and miR-129-2-3p.T).

The mice were killed 3 weeks after viral injection. At high magnification of cerebellar sections from mice expressing GFP-miR-9.T-miR-129-2-3p.T, we found that the PCs were faintly labeled with GFP (lower panel, Supplementary Fig. 10a). This may be because CBh promoter-driven GFP-miR-target expression is strong enough to exceed the capacity of miR-mediated gene silencing. In this study, we electrophysiologically examined whether GFP-miR.T-positive PCs showed functional abnormalities compared to control PCs expressing GFP alone (without miR.T) under the control of the same promoter (upper panel, Supplementary Fig. 10a).

Passive electrical membrane properties of GFP-miR.T-positive PCs (GFP-miRs.T PCs) were similar to the ones of GFP-positive

PCs (control GFP PCs; Supplementary Fig. 10b, left, membrane capacitance, control GFP PCs, 591.6 ± 45.1 pF, *n* = 18; GFP-miR.T PCs, 655.6 ± 56.5 pF, *n* = 14, *t*-test, *P* = 0.377; Supplementary Fig. 10b, right, input resistance, control GFP PCs, 155.2 ± 38.6 MΩ, *n* = 18, GFP-miR.T PCs, 139.7 ± 24.8 MΩ, *n* = 14, *t*-test, *P* = 0.755). We examined spontaneous firing patterns of action potentials (spikes) in PCs by extracellular loose patch recording (Supplementary Fig. 10c, left), and there was no difference in the intervals of spontaneous neuronal spike activities (i.e., spontaneous firing rate) between GFP-miR.T PCs and control GFP PCs (Supplementary Fig. 10c right, control GFP PCs, 43.1 ± 6.6 ms, *n* = 20; GFP-miR.T PCs, 34.7 ± 5.8 ms, *n* = 15, *t*-test, *P* = 0.369). These results suggest that the excessive expression of miR.T has no effect on the intrinsic firing properties of cerebellar PCs.

We also examined synaptic transmission from parallel fibers (PFs) to GFP-miR.T PCs. Basal excitatory synaptic responses at PF-PC synapses were characterized by recording alpha-amino-3-hydroxy-5-methyl-4-i acid (AMPA) receptor-mediated excitatory postsynaptic currents (EPSCs) in response to various stimulus intensities applied to the PFs (Supplementary Fig. 10d, left). There was no significant difference in the stimulus intensity-EPSC amplitude relationship between the GFP-miR.T PCs, and control GFP PCs (Supplementary Fig. 10d, right; multiple *t*-tests

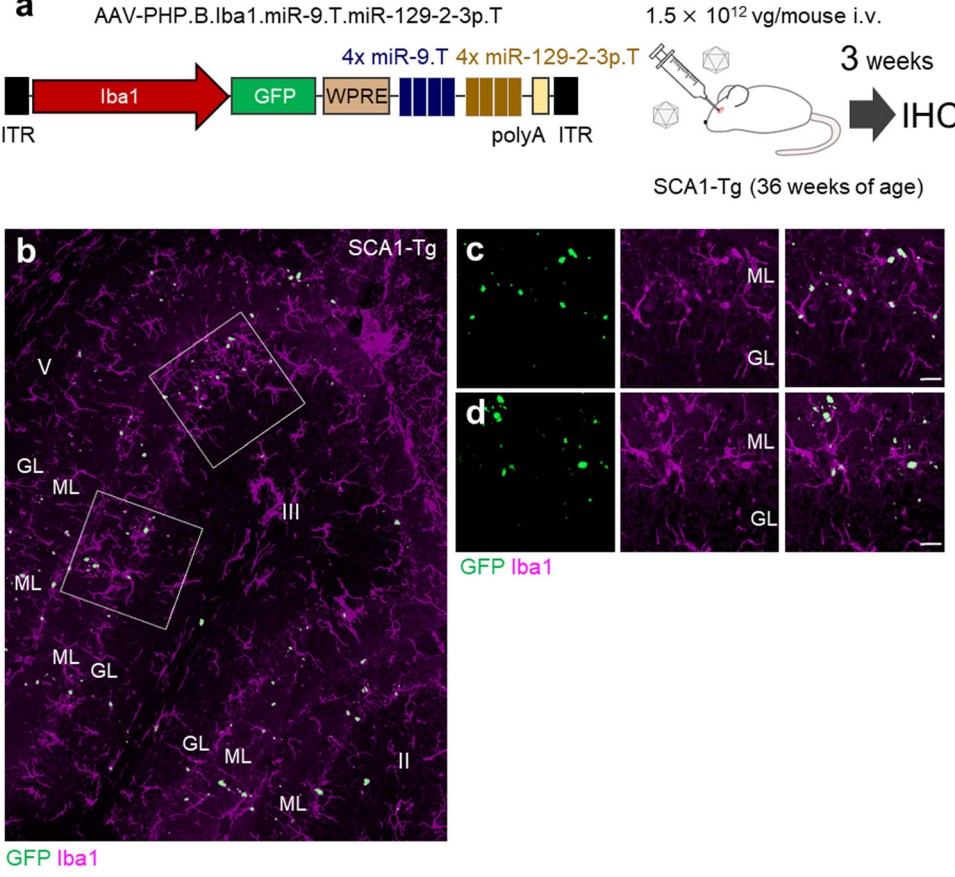

**Fig. 8 Successful microglia targeting, but aggregate formation in SCA1-Tg mice by intravenous administration of AAV-PHP.B. a Diagram showing the AAV construct and experimental procedure.** SCA1-Tg mice at 36 weeks of age received intravenous infusion of AAV-PHP.B.Iba1.miR-9.T.miR-129-2-3p.T (dose: 1.5 ×10$^{12}$ vg/mouse). Three weeks after the viral injection, brain sections were produced and analyzed by double immunolabeling for GFP and Iba1. (**b-d**) Immunofluorescent images of a sagittal section (lobules II and III) from the SCA1-Tg mouse (**b**) and the magnification of square regions (**c**, **d**). Note aggregation of GFP in microglia in SCA1-Tg mouse cerebellum. Scale bars, 20 μm. GL; granule cell layer, i.v.; intravenous injection, ML; molecular layer, SCA1-Tg; spinocerebellar ataxia type 1-transgenic.

corrected with the Holm-Sidak's method, $P > 0.89$ at all intensities). Short-term synaptic plasticity at the PF-PC synapses was examined using paired-pulse stimulation protocols with varying intervals (10 ms to 5 s) between the first and second pulses (Supplementary Fig. 10e, left). The amplitude of the second PF EPSC evoked by a pulse pair was normalized to that of the first PF EPSC, and the ratio (i.e., paired-pulse ratio) was plotted against the inter-stimulus intervals (Supplementary Fig. 10e, right). Both GFP-miR.T PCs and control GFP PCs showed similar synaptic facilitation at shorter intervals (Supplementary Fig. 10e), which is typical of PF-PC synapses[52], and there was no statistical difference in the paired-pulse ratios between GFP-miR. T PCs and control GFP PCs (two-way repeated-measures ANOVA, miR effect, $P = 0.35$). These results suggest that overexpression of miR.T does not influence basal synaptic transmission or short-term synaptic plasticity at the PF-PC synapses.

**GFP aggregation in microglia following intravenous administration of AAV-PHP.B.Iba1.miR-9.T.miR-129-2-3p.T**. Next, we examined whether the systemic application of AAV-PHP.B.Iba1.miR-9.T.miR-129-2-3p.T successfully transduced the microglia in the brain. Wild-type (WT) and SCA1-Tg mice at 36 weeks of age received an intravenous infusion of AAV-PHP.B.Iba1.miR-9.T.miR-129-2-3p.T (1.5 × 10$^{12}$ vg/mouse) and were killed 3 weeks after the injection for immunohistochemistry

(Fig. 8a). Confocal microscopy showed GFP puncta scattered throughout the brain, including the cerebellum and pontine nuclei (WT and SCA1-Tg mice, Fig. 8b and Supplementary Fig. 11).

At higher magnification, we found that the GFP puncta were located in Iba1-immunolabeled microglia (Fig. 8c, d, and Supplementary Fig. 11d, e, g, h). We speculated that this was the lysosomal degradation process of the GFP protein in microglia. To verify this, cerebellar sections were immunolabeled for GFP, Iba1, nuclear DNA (using Hoechst stain), and lysosomal-associated membrane protein 1 (LAMP1), a glycoprotein located across lysosomal membranes[53]. As assumed, GFP aggregates were precisely co-localized with LAMP1 immunoreactivity (Supplementary Fig. 12), indicating that GFP aggregates were present in microglial lysosomes. These results suggest that intravenous administration of AAV-PHP.B.Iba1.miR-9.T.miR-129-2-3p.T causes GFP expression specifically in microglia. However, unlike direct brain tissue injection, GFP is sorted into a lysosomal digestive pathway.

## Discussion

We succeeded in microglia-specific transduction in distinct brain regions of adult mice by direct brain parenchymal injection of AAV vectors carrying the 1.7-kb mouse *Iba1* promoter and two sets of 4-tandem sequence copies, each of which is complementary to different miRNAs, miR-9 and miR-129-2-3p.

The presence or absence and level of endogenous gene expression are strictly regulated in a developmental stage-dependent and cell-type-specific manner. In contrast, AAV-mediated transgene expression using a cell type-specific promoter is less influenced by this biological regulation because of insufficient cell type-specific promoter incorporation, which ranges only a limited genome region, and possible overdose application, leading to ectopic transgene expression in unintended cellular types. In this context, the injection of an optimal dose of AAV carrying a promoter with high specificity to microglia is crucial for microglial targeting. Under our experimental conditions (Fig. 1b, f), the present 1.7-kb mouse *Iba1* promoter worked preferentially in microglia in the striatum (~69%) and cerebellum (~86%), whereas it showed almost no specificity for microglia in the cerebral cortex (~2%), regardless of the lowest dose application (Table 1 and Supplementary Fig. 3), suggesting heterogeneity of brain tissues and region specificity of the 1.7-kb *Iba1* promoter. Another drawback of using viral vectors is the limited packaging capacity, especially in AAV; the relatively long size of the *Iba1* promoter (1.7 kb) restricts the size of the transgene to a maximum of ~2 kb.

Although we took advantage of the *Iba1* promoter for microglia-targeted transgene expression, it should be noted that the *Iba1* promoter is precisely a myeloid cell-specific promoter and works in resident CNS macrophages, including meningeal, perivascular, and choroid plexus macrophages; dendritic cells; and infiltrating monocytes under pathological conditions. Thus, a subtle population of cells transduced with the AAV9.Iba1.miR-9.T.miR-129-2-3p.T may include these cell types.

MicroRNAs guide Argonaute (AGO) proteins and form the targeting module of the miRNA-induced silencing complex (miRISC), which causes translational repression and degradation of targeted mRNAs[54]. Both miR-9 and miR-129-2-3p were shown to be enriched in non-microglial cells but were specifically diminished in microglia[22]. Thus, mRNAs carrying the target sequences are sponged by miRISCs containing miR-9 or miR-129-2-3p, leading to the selective silencing of transgene expression in non-microglial cells. One concern is the simultaneous degradation of miRNAs and their depletion in non-microglial cells including neurons. However, endogenous miR-9 and miR-129-2-3p levels in the three brain regions were not significantly changed, even after overexpression of GFP-miR-9.T-miR-129-2-3p.T using a strong and ubiquitous CBh promoter (Supplementary Fig. 9f), although GFP expression was drastically suppressed (Supplementary Fig. 9d, e). This result is consistent with a recent report[55] showing that fully complementary targets (as in our present case) are cleaved when bound by a catalytic AGO, such as mammalian Ago2, but the pairing of the target mRNA with AGO results in the unloading of the miRNA from AGO and recycling of the miRNA. Consistent with this, motor ability and patch-clamp analyses of the cerebellum were not significantly altered in the mice expressing GFP-miR-9.T-miR-129-2-3p.T throughout the brain (Supplementary Figs. 9 and 10). Nevertheless, impaired neuronal function in other brain regions and association with behavioral defects upon overexpression of miR-9.T and miR-129-2-3p.T cannot be excluded.

In contrast to miR-129-2-3p.T, addition of miR-136-5p.T to AAV9.Iba1.miR-9.T failed to suppress transgene expression in non-microglial cells, despite the expression of miR-136-5p in neurons[21]. A previous study demonstrated that immature dendritic cells expressing low levels of miR-155 did not suppress the expression of a transgene carrying miR-155T, whereas mature dendritic cells that upregulated miR-155 levels induced striking target suppression[32], suggesting that target downregulation occurs at a certain threshold of miRNA expression. This notion is supported by a recent report which revealed that the expression

levels of miR-136-5p in the cerebral cortex were far lower than those of miR-129-2-3p (and miR-9)[51]. Thus, the expression levels of miR-136-5p in non-microglial cells are likely below the threshold for target mRNA degradation.

In addition to ramified microglia in healthy tissues, AAV vectors transduced reactive microglia in the brains of LPS-treated and SCA1-Tg mice. This could be advantageous because our method can be applied to pathological and preclinical experiments targeting neuroinflammation and neurodegeneration, as previously described in Parkinson's disease mouse models[56]. Moreover, the AAV9.Iba1.miR-9.T.miR-129-2-3p.T could be used for functional studies of microglia, as demonstrated by the G-CaMP7.09-based monitoring of intracellular $Ca^{2+}$ mobilization (Fig. 7).

Intravenous application of microglia-targeting AAV-PHP.B resulted in microglia-specific GFP expression. However, unexpectedly, the expressed GFP was transferred to the lysosomal degradation pathway. Since microglia play a role in fighting viruses that enter from peripheral circulation to the brain, the AAV capsid may undergo some modification, such as addition or cleavage of glycosylation, when crossing the BBB to sensitize resident microglia. This result is disappointing for us but could be a partial success in intravenously delivering a transgene product specifically to microglia. Moreover, exploring the mechanism underlying microglial activation by BBB-crossed AAV (but not by AAV directly injected into brain tissues) may lead to the identification of a key interaction (or molecule) that sensitizes resident microglia, which likely facilitates the development of a mutant capsid that does not sensitize microglia.

Collectively, our results validated the efficacy of the direct brain injection of AAV9.Iba1.miR-9.T.miR-129-2-3p.T for the transduction of reactive microglia in a pathological environment as well as ramified microglia in healthy brains. Although microglia targeting the cerebral cortex require careful adjustment of the administration dose and incubation period, the AAV9.Iba1.miR-9.T.miR-129-2-3p.T would be useful for basic research and preclinical studies and may also be promising in clinical studies targeting microglia, since sequences of both miR-9 and miR-129-2-3p are shared broadly from rodents to primates, including Homo sapiens.

During the preparation of this manuscript, AAV capsid variants with high affinity to microglia were reported; however, they were not specific to microglia[57]. The combination of these microglia-tropic mutant capsids with our microglia-targeting Iba1.miR-9.T.miR-129-2-3p.T may help substantially reduce the application dose and attain higher microglial targeting in the cerebral cortex.

## Methods

**Vector construction**. Target sequences were designed based on miRNA sequences obtained from the miRNA Registry (www.mirbase.org). The oligonucleotides used for constructing miR.Ts are presented in Supplementary Table 1. For the PGK.miR-9.T vector, the corresponding sense 1 (S1), sense 2 (S2), antisense 1 (AS1), and antisense 2 (AS2) oligonucleotides were annealed and ligated into the KpnI and BamHI restriction sites in the 3′-UTR of the transgene expression cassette of the parent AAV.PGK vector. For the PGK.miR-9.T.miR-129-2-3p.T or PGK.miR-9.T.miR-136-5p.T vectors and S1, S2, S3, S4, AS1, AS2, AS3, and AS4 oligonucleotides were annealed and ligated into the KpnI and BamHI restriction sites in the 3′-UTR of the GFP expression cassette of the parent AAV.PGK vector.

We used a 1,678-bp fragment from the 5′-flanking region of the first ATG in exon 1 of the mouse *Iba1* (*Aif1*) gene. To clone the *Iba1* promoter, nested PCR was performed using the genomic DNA derived from mouse brain cells. The primer sets Iba1-Nest-F (5′-CCTAGAGCCATCTTGTAAGG-3′) and Iba1-Nest-R (5′-CGAGGAATTGCTGTTGAG-3′) were used for the first amplification of DNA and Iba1-F (5′-ATGCTCTAGACtcgagTACTATAGGATGCATCGTGAAAACC-3′) and Iba1-R (5′-CATGGTGGCGaccggtGGCTCCTCAGACGCTGGTTG-3′) for the second.

For the construction of Iba1, Iba1.miR-9.T, Iba1.miR-9.T.miR-129-2-3p.T (Addgene ID: 190163), or Iba1.miR-9.T.miR-136-5p.T, the PGK promoter in the

corresponding parent AAV.PGK vectors was replaced with the mouse 1.7 kb *Iba1* promoter at the XhoI and AgeI restriction sites.

**Vector production and titration**. AAV2/9 and AAV-PHP.B vectors were produced by co-transfection of HEK293T cells (HCL4517; Thermo Fisher Scientific, Tokyo, Japan) with three plasmids: the expression plasmid, pHelper (Stratagene, La Jolla, CA, USA), and pAAV9.The viral particles were purified using ammonium sulfate or polyethylene glycol 8000 precipitation and iodixanol continuous gradient centrifugation as previously described[58]. The genomic titers of the purified AAV vectors were determined by quantitative real-time PCR using Power SYBR Green Master Mix (Thermo Fisher) with the primers 5′-CTGTTGGGCACTGACAA TTC-3′ and 5′-GAAGGGACGTAGCAGAAGGA-3′ targeting the WPRE sequence. Expression plasmid vectors were used as standards. The concentrated vector expression titer ranged from $6.92 \times 10^{12}$ to $6.30 \times 10^{13}$ vg/mL.

**Animals**. C57BL/6J mice (4–8 weeks old), FVB/N-Tg (Pcp2-ATXN1*82Q) 5Horr mice and their wild-type littermates (24–36 weeks old) were used in this study. In all experiments, careful attention was paid to the sex of the mice to avoid bias toward either males or females. All animal procedures were performed according to protocols approved by the Japanese Act on the Welfare and Management of Animals and Guidelines for Proper Conduct of Animal Experiments, as issued by the Science Council of Japan. The experimental protocol was approved by the Institutional Committee of Gunma University (Nos. 21–063 and 21–065). All efforts were made to minimize suffering and reduce the number of animals used.

**Stereotaxic injections of AAV vectors**. Stereotaxic injections of AAV vectors were performed on 4-week-old C57BL/6J mice, 24-week-old symptomatic FVB/N-Tg mice and their wild-type littermates. Mice were anesthetized by intraperitoneal injection of ketamine (100 mg/kg body weight [BW]) and xylazine (10 mg/kg BW), respectively. Anesthetic depth was monitored using the toe-pinch reflex throughout the surgery, and additional ketamine and xylazine were injected when necessary. A burr hole was created over the injection site to expose the brain. An AAV vector (AAV9.PGK.miR-9.T, AAV9.Iba1, AAV9.Iba1.miR-9.T, AAV9.Iba1.miR-9.T.miR-129-2-3p.T, or AAV9.Iba1.miR-9.T.miR-136-5p.T) was injected into the cerebral cortex, striatum, and cerebellum of mice. To reduce the inoculation volume and increase precision, a 10-μL Hamilton syringe with a 33 G needle was used for the injections. The following stereotaxic coordinates were used for viral injection: cerebral cortex, AP −1.0 mm, ML +1.5 mm, DV +0.9 mm; striatum, AP −1.0 mm, ML +1.75 mm, DV +2.75 mm; cerebellum, AP +6.5 mm, ML 0 mm, DV 2.0 mm (all values are relative to the bregma). The total amount of AAV solution was as follows: cerebral cortex, $1.95 \times 10^9$ vg/mouse; striatum, $3.9 \times 10^9$ vg/mouse; cerebellum, $3.9 \times 10^{10}$ vg/mouse for high-dose injection; cerebral cortex, $5.0 \times 10^8$ vg/mouse; cerebellum, $1.0 \times 10^{10}$ vg/mouse for low-dose injection. The application speeds were as follows: cerebral cortex, 10 nL/min; striatum, 20 nL/min; and cerebellum, 200 nL/min.

**Intravenous injection of AAV-PHP.B**. Intravenous injection of AAV-PHP.B was carried out on 36-week-old symptomatic FVB/N-Tg mice and their wild-type littermates. After deep anesthesia, 100 μL of AAV-PHP.B solution ($1.5 \times 10^{13}$ vg/mL) was intravenously injected into the retro-orbital sinus of mice using a 0.5 ml syringe with a 30-gauge needle (08277; Nipro, Osaka, Japan).

**Immunohistochemistry**. Mice were transcardially perfused with 4% paraformaldehyde in 0.1 M PB (pH 7.4), the brain was post-fixed for 6–8 h, and transferred to 1x PBS. The brains were cut into 50-μm coronal slices for the cerebral cortex, striatum, and sagittal slices of the cerebellum using a microtome (VT1000S or VT1200S; Leica, Wetzlar, Germany). Slices were treated with a blocking solution (5% normal donkey serum, 0.5% Triton X-100, and 0.05% NaN₃ in PBS) for 1 h and incubated overnight at 4 °C with the primary antibodies summarized in Supplementary Table 2. After washing with PBS three times for 15 min each, slices were incubated 2 h to overnight at 4 °C with the secondary antibodies summarized in Supplementary Table 2. Finally, the slices were rinsed in PBS three times, incubated with Hoechst 33342 nucleic acid stain (Invitrogen catalog number H1399), mounted in ddH₂O, mounted in ProLong Diamond/Glass Antifade Mountant (Thermo Fisher), and stored at 4 °C in the dark. Immunofluorescence was analyzed using a Zeiss LSM 800 laser-scanning confocal microscope and accompanying software (Carl Zeiss, Oberkochen, Germany).

**Confocal microscopy analysis**. The proportion of brain myeloid cells transduced with AAV vectors was examined by immunohistochemistry. The 50-μm-thick tissue sections obtained from the cerebral cortex, striatum, and cerebellum were immunostained for GFP and Iba1. Thirty consecutive optical sections at 0.5 μm intervals (25,513.67 μm², total depth of 15 μm) were captured using a Zeiss LSM 800 microscope with a 40x/0.95 NA objective, and images were subsequently processed with Zeiss Zen software to create a maximum intensity projection (MIP) image. From each brain slice, 32 MIP images were acquired without overlapping (total 816,386.42 μm²) and the number of GFP-expressing myeloid cells was counted. GFP-positive cells co-labeled with Iba1 were considered to be brain myeloid cells. Three brain slices per mouse and 4–9 mice were analyzed. A minimum of 500 cells were counted.

**Confocal live Ca²⁺ or GFP imaging in microglia**. Confocal Ca²⁺ or GFP imaging of acute cerebellar slices was performed as described previously[27,39], with some modifications. Parasagittal slices (200-250 μm in thickness) of the cerebellar vermis were prepared using a vibroslicer (VT1200S; Leica, Germany) and maintained in a solution (ACSF) containing (in mM) 125 NaCl, 2.5 KCl, 2 CaCl₂, 1,0 MgCl₂, 1.25 NaH₂PO₄, 26 NaHCO₃, and 20 D-glucose, and bubbled with 95% O₂ and 5% CO₂ at room temperature for more than 1 h before the beginning of recordings. Image processing and analysis were performed using Andor iQ2 (Andor), NIH ImageJ, Igor Pro8 (WaveMetrics), and custom-written programs by NH.

G-CaMP7.09 is a recently developed gene-encoded Ca²⁺ indicator that increases its fluorescence when the intracellular Ca²⁺ concentration is elevated[43]. To record Ca²⁺ signals from G-CaMP7.09-expressing cells, confocal fluorescence images were acquired every 2 s (200–300 ms exposure time, 512 × 512 pixels, no binning) with a ×40 water immersion objective (LUMPLFLN 40XW; Olympus, Tokyo, Japan), a water-cooled CCD camera (iXon3 DU-897E-CS0-#BV-500; Andor, Belfast, Northern Ireland), and a high-speed spinning-disk confocal unit (CSU-X1; Yokogawa Electric, Tokyo, Japan) attached to an upright microscope (BX51WI; Olympus, Tokyo, Japan). A 488-nm light beam from a diode laser module (Stradus 488-50; VORTRAN, Sacramento, CA) was used for excitation, and the emitted fluorescence was collected through a band-pass (500–550 nm) filter. During recordings, cerebellar slices were perfused with an ACSF bath solution at room temperature.

To evoke an intracellular Ca²⁺ increase in microglia, 100 μM ATP dissolved in the ACSF bath solution was applied extracellularly via a gravity-fed bath-application device[44].

Because single continuous recordings lasted for more than 10 min, image drift (translation drift) was occasionally observed. In these cases, drifted images were corrected using the Image Stabilizer plugin in ImageJ (http://www.cs.cmu.edu/~kangli/code/Image_Stabilizer.html). G-CaMP fluorescence at time t ($F_t$) in each pixel was background-subtracted, and the Ca²⁺-dependent relative increase in fluorescence was measured by calculating $\Delta F/F_{basal}$, where $F_{basal}$ is the basal fluorescence intensity averaged during pre-stimulus frames (i.e., frames before ATP application), and $\Delta F = F_t - F_{basal}$. Background fluorescence was obtained from a region lacking the cell structure in the same frame. The mean $\Delta F/F_{basal}$ values in each region of interest (ROI) were calculated for each frame. ROIs were placed on G-CaMP-positive cellular structures. We could not determine whether multiple adjacent ROIs in a frame were on the same cell or on other different cells; thus, we referred to a single ROI as a microglial cellular compartment. Usually, coverage of a microglial process territory reaches within a few hundred micrometer[42], and Ca²⁺ imaging data in this study were collected from 12 different fields of view, more than 300 μm apart, in five slices of two mice. Therefore, we can safely say that our Ca²⁺ imaging data set came from more than 12 microglia. To quantify ATP-induced Ca²⁺ signals in G-CaMP positive cells, the peak amplitude of $\Delta F/F_{basal}$ was measured in a time window of 120 s after ATP application onset.

To record live morphological dynamics in GFP-positive cells, confocal GFP fluorescence images were acquired every 2–6 s (200-250 ms exposure, 512 × 512 pixels, no binning). The signal-to-noise ratio of GFP signals was much higher than that of G-CaMP signals under our experimental conditions. Image drift in GFP imaging was corrected comparably to that in Ca²⁺ imaging.

**Rotarod and beam-walking test**. To examine whether the behavioral phenotype was affected by overexpression of miRNA target sequences, 6- to 7-week-old C57BL/6J mice were intravenously injected with AAV-PHP.B.CBh-GFP (PHP.B.CBh-GFP), or AAV-PHP.B.CBh-GFP-miR-9.T.miR-129-2-3p.T (PHP.B.CBh-GFP-miR.T), respectively at a titer of $6 \times 10^{11}$ vg. Three weeks later, motor performance was evaluated using the rotarod test on a treadmill (MK-610A/RKZ; Muromachi Kikai, Tokyo, Japan). The mice were subjected to three trials at 30-min intervals. The rotation speed was accelerated from 4 to 50 rpm in 300 s, and the time taken to fall off the rod was measured. After the rotarod test, the mice were further tested using a beam-walking test. After habituation, the mice were placed on one edge of a bar (600 mm long and 11 mm in diameter) and walked on the bar three times. The performance of the mice walking on the beam was recorded using a digital video camera (shutter speed:1/500, 60 fps; HC-VX985M; Panasonic, Osaka, Japan) and the average time for the mice to walk through a 600-mm-long beam was measured.

**Endogenous miRNA measurement**. To measure the number of endogenous miRNAs, 8-week-old C57BL/6J mice were intravenously injected with either PHP.B.CBh-GFP or PHP.B.CBh-GFP-miR. T, at a titer of $6 \times 10^{11}$ vg/mouse. Three weeks after injection, the mice were deeply anesthetized and perfused with cold PBS (-). The brains were then quickly extracted and divided into two parts, with the left-right axis along the midline. The left brains (for histological observation) were immersed in cold 4% paraformaldehyde (PFA) overnight at 4 °C and replaced with 4% PFA with 1 × PBS (-) the next day. Sagittal sections (50 μm thickness) were then produced using a microtome VT1200S (Leica). Slices were fluorescently immunostained with anti-GFP (04404-84; Nacalai Tesque) and anti-NeuN (MAB377;

Merck) antibodies and mounted using ProLong Diamond Antifade (Thermo Fisher Scientific). Photomicrographs of immunostained sections were obtained using a BZ-X800 fluorescence microscope (Keyence). The right half of the brain was used to measure the miRNA levels. The parietal lobes of the cerebral cortex, cerebellum, and striatum were collected and placed separately using the TRIzol reagent (Thermo Fisher Scientific). The tissue was homogenized using BioMasher II (320103; Nippi, Tokyo, Japan) and Power Masher II (891300; Nippi), and rapidly quenched with liquid nitrogen. The tissues were stored at −80 °C until RNA was collected. The miRNeasy Mini Kit (217004; QIAGEN, Hilden, Germany) was used to collect total RNAs containing the miRNAs. Total RNA was reverse-transcribed using the TaqMan MicroRNA Reverse Transcription Kit (4366596; Thermo Fisher Scientific) and miRNA-specific reverse transcription primers (Assay ID:000583; 4427975; Thermo Fisher Scientific). The amounts of miR-9, miR-129-2-3p, and miR-129-5p in the reverse-transcribed miRNA samples were quantified using a TaqMan probe (Assay ID:000583; 4427975; Thermo Fisher Scientific) and TaqMan Fast Advanced Master Mix (4444556; Thermo Fisher Scientific) in a thermal cycler for real-time PCR (Thermal Cycler Dice Real-Time System II, Takara Bio, Kusatsu, Japan).

**Electrophysiological analysis of cerebellar PCs overexpressing GFP or GFP-miR.T**. Acute cerebellar slices were prepared and maintained as described in the Methods section on 'Confocal live imaging' below. Whole-cell patch-clamp recordings were performed using a MultiClamp 700 B amplifier (Molecular Devices, USA) in voltage-clamp mode (holding potential of -70 mV) at room temperature from the somata of cerebellar PCs using patch pipettes (1–4 MΩ) pulled from borosilicate glass (Harvard Apparatus) or #0010 glass (PG10165-4, World Precision Instruments). The pipette solution contained (135 mM) 135 K-gluconate, 10 mM HEPES, 5 mM KCl, 5 mM NaCl, 5 Mg-ATP, 0.5 Na-GTP, 0.1 mM EGTA, and 5 mM phosphocreatine (pH 7.3). The liquid junction potential is not corrected. The passive electrical membrane properties of the recorded PCs were estimated using the averaged traces of 20 current responses evoked by hyperpolarizing voltage pulses (from −70 to −75 mV, 500 ms duration) in the voltage-clamp mode. To record parallel fiber (PF)-evoked EPSCs, the ACSF extracellular solution (see 'Confocal live imaging' section below) was supplemented with picrotoxin (100 μM) to block GABA$_A$ receptor-mediated inhibitory synaptic currents, and PFs were stimulated by applying square pulses (60 μs, 10–60 μA) through a glass pipette filled with the extracellular solution and placed in the molecular layer of the cerebellar slice. For extracellular loose patch recording, the patch pipettes were filled with extracellular solution and placed close to the somatas of the GFP-positive PCs (loose cell-attached). Spontaneous action currents (generated by spontaneous neuronal firing) were recorded using an EPC-8 amplifier (HEKA Elektronik GmbH, Germany) for at least 1 min at a pipette potential of 0 mV. Spike occurrence was detected using current-level thresholding. When the waveforms of the detected action currents were not uniform, data were discarded. The inter-spike intervals of all detected events were measured in each PC, and an average of the intervals was taken as a value representative of each recorded PC. All electrophysiological data were acquired using the pCLAMP10 software (Molecular Devices) and analyzed using Igor Pro 8 (Wavemetrics) with Neuromatic (http://www.neuromatic.thinkrandom.com/)[59] and custom-written Igor procedures by NH.

**Statistics and reproducibility**. Statistical analyses were performed using IBM SPSS Statistics 22 (IBM Corp., Armonk, NY) and GraphPad Prism 8 (GraphPad Software Inc., San Diego, CA). All data were checked for compliance with statistical assumptions for each test, including normal distribution and equal variances across groups. Unless otherwise indicated, data are presented as mean ± SEM and two-tailed unpaired $t$-tests or ANOVA multiple comparison tests were used to assess the statistical significance. Quantitative data of microglial specificity in the cerebellum one week after viral injection were analyzed by nonparametric Kruskal–Wallis test because data normality was not confirmed, and a post hoc Dunn's test with Bonferroni adjustment for multiple comparisons was used to determine the statistical significance (Table 1). Electrophysiological data of PF EPSC amplitude and paired-pulse ratio were analyzed using multiple $t$-tests corrected with Holm-Sidak's method and two-way repeated-measures ANOVA, respectively (Supplementary Fig. 10d, e). Statistical significance was considered at $P < 0.05$ (or equivalent corrected for multiple comparisons). No statistical methods were used to predetermine the sample sizes, but our sample sizes were similar to those reported in previous studies. In graphs for electrophysiological experiments, bar graphs representing mean values and the individual data points are displayed in parallel. In boxplots, centerlines, bonds of box, and whiskers represent the median, 25th/75th percentiles, and minimum/maximum values, respectively. Further details of statistical analysis are reported in the figure legends.

**Reporting summary**. Further information on research design is available in the Nature Portfolio Reporting Summary linked to this article.

## Data availability
The Plasmid containing the Iba1 promoter and two types of miR.T used in this study are available from Addgene (Addgene ID: 190163). Microglial morphological changes associated with this study are present in the paper and Supplementary Movies. The source data for each related graphs is provided in the Supplementary Data 1 file. The data supporting the findings of this study are available from the corresponding author H.H. upon reasonable request.

## Code availability
All custom-written analysis codes are available from the corresponding author upon request.

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

## Acknowledgements

The authors thank Asako Ohnishi, Nobue McCullough, and Ayako Sugimoto for the production of the AAV vectors, Junko Sugiyama for raising the mice, and Junko Sugi for some mice experiments. This research was partially supported by the program for Brain Mapping by Integrated Neurotechnologies for Disease Studies (Brain/MINDS) from the Japan Agency for Medical Research and Development (AMED) under Grant Numbers JP19dm0207057, JP20dm0207057, JP21dm0207111, and JSPS KAKENHI (Grant Numbers 15H04254, 16K15477, 18H02521, 15K18330, 19K06899, and 22K06454).

## Author contributions

Y.O., A.K., and H.H. designed the experiments; Y.O., K.N., Y.S., Y.M., Y.F., A.H., N.H., and A.K. performed the research; N.H. conducted the $Ca^{2+}$ imaging experiments; J.N. provided the experimental material; Y.O. drafted, and H.H. completed the study.

## Competing interests

The authors declare no competing interests.
