## [Peer Review File · Communications Biology]

Reviewers' comments:

Reviewer #1 (Remarks to the Author):

In this report, Okada Y et al created microglia (MG)-targeting AAV vectors containing a 1.7-kb promoter of Iba1 gene, one of the MG biomarkers. They also included a repeated miRNA target sites for microRNA (miR)-9 and miR-129-2-3p, which markedly suppressed transgene expression in non-MG cells. Interestingly, AAV vectors transduced ramified MG not only in healthy tissues but also in MG in mice treated with LPS and in a mouse model of neurodegenerative disease. This study reported an interesting MG-targeting AAV vector, which can be utilized to study microglial pathophysiology and MG-targeted therapies, especially in the regions of striatum and cerebellar cortex. I believe these findings are important and suitable to be published in the journal.

Minor points:

1. The overall promoter region is relatively large (>1600bp) while AAV titers were relatively high. It is not clear how the authors plan to improve the delivery efficiency to MG for their future work in the pre-clinic/clinic setting.
2. Across the study, while IBA1 staining was useful as a biomarker for microglia, co-localization of GFP with other new/specific MG (TMEM119 or HexB) marker may be tested. Astrocyte (GFAP) specific marker may be also included to further confirm the specific MG targeting.
3. In Fig 3g, dosage label was confusing, it should include either μ l or MOI of the viruses.
4. In Fig 4, how about 3 week after LPS administration?
5. Line 225, it should be fig 5a not 7a.
6. Fig 5, how about 1 week after infection? Is it similar to 3 week-infection regarding MG targeting?
7. From Fig 9 to Fig 11, are any quantitation data available for GFP+/Iba+ cells?
8. Can the authors provide some evidence to support that these MG-targeting tools can be apply in human MG, not just mouse MG?

Reviewer #2 (Remarks to the Author):

In this manuscript, authors describe and characterize an adeno-associated virus (AAV) vector that targets microglia in brain. Iba 1 promoter is used to limit transgene expression to microglia in addition to miR-9.T and miR-129-2-3p.T. All modifications are performed in the AAV genome as transgene cargo and the optimized gene delivery is independent of the AAV serotype. Authors show that microRNAs, miR-9.T and miR-129-2-3p.T, suppress the expression of the delivered genes in non-microglial cells of mouse brain.

Authors also investigated the application of the improved gene delivery by their vector in LPS-treated and SCA1-Tg mice. No significant microRNA overexpression or electrophysiological impairments were observed due to overexpression of miR-9.T and miR-129-2-3p.T.

Direct stereotactic injection of the AAV9-Iba1.miR-9T.miR-129-2-3p.T into the brain led to specific and robust gene expression in microglia of striatum, cerebellum and cerebral cortex. However, retroorbital injection of AAV-PHP.B-Iba1.miR-9T.miR-129-2-3p.T resulted in delivery of the transgene to lysosomes of microglia.

Overall, this is a very-well organized and comprehensive study of the AAV vector attributes that enhance gene delivery to microglia. A lot of effort has been dedicated to gathering and analyzing data. Experiments have been rationally designed and presented data supports authors claims. The characterized vector attributes and described methods and doses will contribute significantly to successful gene delivery to microglia.

AAV variants can be engineered for specific animal model to target microglia. The findings in this study can be applied to vectors with any serotype to ensure preferential expression in microglia.

Although, gene delivery with the AAV-PHP.B serotype was not successful, the data shown in this study will save readers time and effort by cautioning against using PHP.B serotype with this vector (or probably any other vector) to target microglia.

AAV particles can accommodate up to 4.2kb of genetic material between their ITRs. Any larger cargo size will give rise to virions with partial cargo. The promoter region used in this vector is 1.7kb which limits the size of the rest of the transgene (discussed briefly in discussion). Still, this is a very useful vector when used in stereotactic injections.

This study contributes to improved transgene delivery into the brain microglia which will be beneficial for neuroinflammatory disease research. This is a significant contribution to the field and I highly recommend the publication of this article.

Minor suggestions:

1. Please specify the AAV serotype in text and figures - for example, AAV9.Iba1.miR-129-2-3p.T.
2. Please add a description of AAV serotypes with improved microglia targeting to your introduction (discuss pros cons).

Reviewer #3 (Remarks to the Author):

The manuscript addresses the question on how to efficiently transduce microglia with adeno-associated virus (AAV) to allow manipulation of these cells. The authors show that they can efficiently transduce microglia with GFP as well as can record calcium dynamics of GCamp-transduced microglia. For their AAV, the authors take advantage of first, Iba1 as a microglia-selective promoter and, second, the expression of the microRNA(miR)-9 and 129-2-3p, which suppresses transgene expression in non-microglial cells. This point is the main limitation of the strategy, which will make it unlikely to be translated into clinics (L433). MicroRNAs can have hundreds of target mRNAs and have a multiplex role in influencing mRNA translation. Therefore, the expression of mRNAs like miR-9 will not only downregulate the undesired transgene but might also sequester mRNAs from the host cell. The authors take this point serious and performed experiments that started to address this aspect including behavior analysis and electrophysiological recordings in purkinje cells.

There are two interesting, so far unknown, observations in the manuscript that makes this paper interesting for future follow-up studies: First, that even independent of the injected AAV dosage, the Iba1 promoter is more specific in transducing microglia in the striatum and the cerebellum but not in the cortex (Supplementary Figure 6). Such a region-specificity has not been described. Second, when they peripheral injected their microglia-targeting AAV, they find that the virus is capable to cross the blood-brain barrier. Interestingly, the AAV ends up in the microglia lysosome (Supplementary Figure 10). This is not happening when they directly inject the AAV into the parenchymal, which suggests modifications on the capsid that might cause the AAV to be recognized and be taken up (as the author suggested in L420-427). Both results can provide new perspectives to further refine AAV transduction strategies and provide original insights. It might be advisable to transfer Supp. Fig. 6 and 10 to the main figures, as they provide new insights that are not known.

Overall, this manuscript would be interesting for the microglia community and virologists that are interested in refining microglia-targeting strategies. Please find below some remarks to strengthen the manuscript.

Additional point to address:

- 1.) As a functional readout that the overexpression of miR-9 and miR-129-2-3p in non-targeted cells does not cause a side-effect, the authors used the AAV-PHP.B that is injected in the periphery, passes the blood-brain barrier, and targets the cells in the entire brain. They have chosen two behavioral tests related to motor function, which only represents a snapshot of the behavioral repertoire (L273-

275). Therefore, the authors should provide a rationale for the choice of these behavior tests in relation to the impact of miR-9 and miR-129-2-3p and discuss the limitations of these tests. The same holds true for the recording of Purkinje cells (PC). The microRNAs might not have an effect in the PC but they influence the function in neurons of other brain regions. Therefore, the statement in L336-338 should be more precisely phrased.

2.) The authors should consider to share their plasmid constructs in a public available repository such as addgene.com or plasmids.eu to provide the community fast access to this tool

Minor comments

3.) Certain sentences require some clarity in the English to avoid contradictive statements and ambiguities:

- a) L44-46: outlines the diverse microglia function but then L46-47, the exact functions are not known.
- b) L56-L61: outlines reasons why macrophages are not possible to transduce such as that they are trained to fight viral transduction. But then in the next sentence (L61-64), this reason does not hold.
- c) L67: The content in the bracket causes confusion because in the sentence before, it has been stated that miR-9 is absent of resident microglia. How can something that is being absent be lost?
- d) L141: It is not clear to what the "not more than five times" is referring to. The authors have chosen a 4-time repeat, so it is clear that this is not 5-time.
- e) To avoid confusion between the terminologies "cerebellar" and "cerebral", the authors should consistently use cerebellum. This would also match their figure annotations.

4.) Typo: L64: "they" in capital letter.

5.) L86: Is there a rationale for choosing these three brain regions? The authors should also clarify, whether they have targeted a defined cortical brain region.

6.) L288: miR.T and L290: GFP-miR.T should be defined also in the text and not only in Supplementary Figure 7a.

7.) L342-342: a "whether" is missing in this sentence?

8.) Either the Supplementary videos are not in the order as they are cited in the manuscript, or the figure legends of the videos 1-5 are not correctly referencing to the relevant figures.

9.) Supplementary Figure 8 misses the statistics between the conditions.

Answers to the comments from reviewer #1

We are grateful to reviewer #1 for the critical comments and useful suggestions that have helped us to significantly improve our paper. As indicated in the responses that follow, we have taken all these comments and suggestions into account in the revised version of our paper. Our point to point responses were depicted in blue.

Comment #1. *The overall promoter region is relatively large (>1600bp) while AAV titers were relatively high. It is not clear how the authors plan to improve the delivery efficiency to MG for their future work in the pre-clinic/clinic setting.*

Response

Originally, we had an idea of using Tet-off system, which injects mixture of two AAVs, one expressing tTA by the Iba1 promoter (the present AAV vector), and another one expressing a therapeutic gene under the control of TRE promoter. Recently, AAV capsid variants with high affinity to microglia were reported, although they are not specific to microglia (Lin R. et al., Nat Methods. 2022 Aug;19(8):976-985). Combination of those microglia-tropic mutant capsids with our microglia-targeting Iba1.miR-9.T.miR-129-2-3p.T may help substantially reduce application dose, and attain higher microglia targeting also in the cerebral cortex.

This was discussed in the text (Page 22, line 452-456).

Comment #2. *Across the study, while IBA1 staining was useful as a biomarker for microglia, co-localization of GFP with other new/specific MG (TMEM119 or HexB) marker may be tested. Astrocyte (GFAP) specific marker may be also included to further confirm the specific MG targeting.*

Response

We purchased two anti-TMEM119 antibodies, one of which worked well in immunohistochemistry. We examined the striatum and cerebellum, and found that all GFP- and Iba1-double-positive cells were labeled also with TMEM119 (Suppl. Fig. 7). In addition, we performed immunohistochemistry using anti-GFAP antibody, and confirmed absence of GFP- and GFAP-double-positive cells (astrocytes) in the striatum and

cerebellum (Suppl. Fig. 8). These results were described in the text (Page 9-10, line 181-198).

Comment #3. *In Fig 3g, dosage label was confusing, it should include either μ l or MOI of the viruses.*

Response

Viral doses injected to mice were described above the schemas (Fig. 3b, c, e). In addition, the dosage label “Dose” in Fig. 3g was replaced by “Relative dose” in new Fig. 4g.

Comment #4. *In Fig 4, how about 3 week after LPS administration?*

Response

We appreciate this reviewer for the critical comment. This LPS experiment was aimed to see whether our AAV vector works also in activated microglia. Therefore, we used high dose (3.9×10^{10} vg/ml), and assessed the cerebellar section in one week (just after 7-day consecutive LPS injection). Since we used high dose, some non-microglial cells may be transduced after three weeks as shown in Fig. 3g. In addition, since reactive microglia return to ramified microglia three weeks after the injection (two weeks after cessation of the LPS injection), we did not examine cerebellar sections three weeks after LPS treatment.

Comment #5. *Line 225, it should be fig 5a not 7a.*

Response

We are sorry for the inadvertent mistake. We corrected it (line 244), and checked all figure numbers throughout the manuscript.

Comment #6. *Fig 5 (SCA1-Tg mouse), how about 1 week after infection? Is it similar to 3 week-infection regarding MG targeting?*

Response

It is supposed that lower number of microglia were labeled by GFP in one week after injection, because expression levels of a transgene continue to increase over 3 weeks. In this experiment, we injected low dose AAV (1.0×10^{12} vg/mL, 10 μ L) to SCA1-Tg mouse cerebellum. To obtain robust and efficient labeling of microglia, we waited for three weeks.

Comment #7. *From Supplementary Fig 9 to Fig 11, are any quantitation data available for GFP+/Iba+ cells?*

Response

All aggregates were localized in Iba1-labeled microglia. We thought that some aggregates in microglia may have digested and disappeared at the time of examination. Therefore, we did not perform quantification.

Comment #8. *Can the authors provide some evidence to support that these MG-targeting tools can be apply in human MG, not just mouse MG?*

Response

This is really a critical point. As discussed in the text, sequences of both miR-9 and miR-129-2-3p are shared broadly from rodents to human (line 450-451), and thus, miR-T sequences are supposed to work also in human brain. As for the Iba1 promoter, we are not sure whether it works also in human. We are planning to test our microglia-targeting AAV in mixed culture containing iPS cell-derived human microglia (iCell[®] Microglia, Fujifilm CDI). If our AAV vectors do not work in human, we will replace mouse Iba1 promoter sequence with that of human Iba1 promoter using the homologous genome region. However, these experiments may be beyond the scope of this study, and we would like to use the results in a next paper.

Answers to the comments from reviewer #2

We are grateful to reviewer #2 for the helpful comments and suggestions. As indicated in the responses that follow, we have taken all these comments and suggestions into

account in the revised version of our paper. Our point to point responses were depicted in blue.

Comment #1. *Please specify the AAV serotype in text and figures - for example, AAV9.Iba1.miR-129-2-3p.T.*

Response

We specified the AAV serotypes throughout the text and figures.

Comment #2. *Please add a description of AAV serotypes with improved microglia targeting to your introduction (discuss pros cons).*

Response

Along with previous studies, we added a description of AAV serotype for improved microglia targeting (Page 4, line 63-71).

Answers to the comments from reviewer #3

We are really grateful to reviewer #3 for the critical comments and useful suggestions that have helped us to significantly improve our paper. We have taken all these comments and suggestions into account in the revised version of our paper. Our point to point responses were depicted in blue.

Comment #1. *It might be advisable to transfer Supp. Fig. 6 and 10 to the main figures. Both results can provide new perspectives to further refine AAV transduction strategies and provide original insights.*

[There are two interesting, so far unknown, observations in the manuscript that makes this paper interesting for future follow-up studies: First, that even independent of the injected AAV dosage, the Iba1 promoter is more specific in transducing microglia in the striatum and the cerebellum but not in the cortex (Supplementary Figure 6). Such a region-specificity has not been described.]

Response

We agree with this reviewer's thoughtful comment. Supp. Fig. 6 shows transduction efficiency of microglia after our AAV injection. Therefore, we prepared a new graph showing the transduction specificity of microglia by AAV vectors in three distinct brain regions (below), and compared with Table containing essentially the same data (below).

Table 1. Microglia specificity (GFP- and Iba1-positive cells / GFP (+) cells)

AAV construct		Cerebral cortex	Striatum	Cerebellum	n
Promoter	miR target				
PGK	miR-9.T	10.2 ± 3.7 (319/2656)	7.4 ± 3.4 (191/2576)	34.4 ± 10.8 (901/2620)	5
Iba1		2.1 ± 1.3* (59/2791)	68.8 ± 9.6** (1808/2628)	85.7 ± 5.5 (2244/2618)	5
Iba1	miR-9.T	27.3 ± 2.4**†† (753/2760)	94.0 ± 2.0**†† (2663/2833)	100 ± 0*** (2726/2726)	5
Iba1	miR-9.T miR-129-2-3p.T	86.5 ± 5.6**†††† (2750/3180)	99.6 ± 0.5**†† (3127/3140)	100 ± 0*** (2712/2712)	5

Data are presented as mean ± s.d. (%). The numbers in the rightmost row (n) indicate the number of mice examined. The numbers in parentheses indicate the number of cells counted (GFP- and Iba1-positive cells/GFP-positive cells). * vs PGK.miR-9.T, † vs. Iba1, ‡ vs. Iba1-miR-9. AAV; adeno-associated virus, Iba1; ionized calcium-binding adaptor molecule 1, miR; microRNA, n; number of mice, PGK; phosphoglycerate kinase 1.

Both look fine, but Table contains more information. So, we decided to use Table.
In addition, Supp. Fig. 10 was transferred to main figure (new Fig. 8).

Comment #2. *As a functional readout that the overexpression of miR-9 and miR-129-2-3p in non-targeted cells does not cause a side-effect, the authors used the AAV-PHP.B that is injected in the periphery, passes the blood-brain barrier, and targets the cells in the entire brain. They have chosen two behavioral tests related to motor function, which only represents a snapshot of the behavioral repertoire (L273-275). Therefore, the authors should provide a rationale for the choice of these behavior tests in relation to the impact of miR-9 and miR-129-2-3p and discuss the limitations of these tests.*

Response

We assessed motor ability because the cerebellum expressed higher levels of miR-9 and miR-129-2-3p than other brain regions such as the cerebral cortex, brainstem, and spinal cord (Pomper et al. 2020). This was described in the text (Page 14, line 290-294).

Comment #3. *The same holds true for the recording of Purkinje cells (PC). The microRNAs might not have an effect in the PC but they influence the function in neurons of other brain regions. Therefore, the statement in L336-338 should be more precisely phrased.*

Response

I appreciate this reviewer's critical comment. The original statement in Results section was removed, and rephrased in the Discussion section (Page 20, line 416-417).

"Nevertheless, impaired neuronal function in other brain regions and associated behavioral defects by overexpression of miR-9.T and miR-129-2-3p.T cannot be excluded."

Comment #4. *The authors should consider to share their plasmid constructs in a public available repository such as addgene.com or plasmids.eu to provide the community fast access to this tool.*

Response

We have deposited the plasmids in Addgene to make it available to the community (Addgene ID: 190163, Page 23, line 477-478). The plasmid will be available after publication of this paper.

Minor comments

Comment #5. *Certain sentences require some clarity in the English to avoid contradictive statements and ambiguities:*

a) L44-46: *outlines the diverse microglia function but then L46-47, the exact functions are not known.*

Response: The sentence (*Nonetheless, their exact functions in the CNS have not yet been fully elucidated.*) was removed to solve the contradiction (Page 3, line 43).

b) L56-L61: *outlines reasons why macrophages are not possible to transduce such as that they are trained to fight viral transduction. But then in the next sentence (L61-64), this reason does not hold.*

Response: The sentence (microglia are brain macrophages trained to fight viral infection) was removed, and the section was revised for clarity (Page 4, line 63-68) (below).

“They however resulted in only minor success with some “single microglia targeting,” probably because of weak promoter activity and the low binding capacity of AAV capsids to microglia. Thus, for the efficient and selective transduction of microglia, a robust microglia-specific promoter and microglia-tropic AAV capsid are required.”

c) L67: *The content in the bracket causes confusion because in the sentence before, it has been stated that miR-9 is absent of resident microglia. How can something that is being absent be lost?*

Response: “loss” was used to show “absence”. But, for clarity, this phrase (because of loss of miR-9) was removed (Page 3, line 54-57).

d) L141: *It is not clear to what the “not more than five times” is referring to. The authors have chosen a 4-time repeat, so it is clear that this is not 5-time.*

Response: “not more than five times” was removed for clarity (Page 7, line 142-143).

e) *To avoid confusion between the terminologies “cerebellar” and “cerebral”, the authors should consistently use cerebellum. This would also match their figure annotations.*

Response: “cerebellar cortex” was replaced by “cerebellum” throughout the text.

4.) *Typo: L64: “they” in capital letter.*

Response: Corrected.

5.) *L86: Is there a rationale for choosing these three brain regions? The authors should also clarify, whether they have targeted a defined cortical brain region.*

Response: We added the following sentences (Page 5, line 83-88):

“the cerebral cortex (frontal cortex), striatum, and cerebellum. The striatum was selected to compare our results with a previous report using lentiviral vectors with miR-9T, while along with the striatum, the frontal cortex and cerebellum were chosen as aberrant microglial function in these brain regions have been associated with various neurodegenerative disorders, such as Alzheimer’s disease and cerebellar ataxias.”

6.) *L288: miR.T and L290: GFP-miR.T should be defined also in the text and not only in Supplementary Figure 7a.*

Response: Revised.

7.) *L342-342: a “whether” is missing in this sentence?*

Response: Corrected.

8.) *Either the Supplementary videos are not in the order as they are cited in the manuscript, or the figure legends of the videos 1-5 are not correctly referencing to the relevant figures.*

Response: We checked the order of Suppl. Videos carefully and corrected.

9.) *Supplementary Figure 8 misses the statistics between the conditions.*

Response: The detailed statics were described in the text (Page 16, line 327-354). There are no statistically significant differences between groups in Suppl. Fig. 10 (previous Suppl. Fig. 8).

REVIEWERS' COMMENTS:

Reviewer #1 (Remarks to the Author):

I thank the Authors for their new work to address my comments and concerns, which makes this manuscript better to be published in the journal.

Reviewer #2 (Remarks to the Author):

This manuscript was previously reviewed by me and two other reviewers. Authors have made all appropriate changes to address reviewer's comments and requests. This manuscript significantly contributes to the field and I highly recommend its publication.

Reviewer #3 (Remarks to the Author):

The authors addressed my comments and I do not have any further scientific comments.

Recommendation: It might be advisable to have an English editorial proof reading to further improve the clarity of certain sentence structures (e.g.

L54-57 the terms "expressed" and "non-microglial cells" are repeated 3x or

L261-262: the sentence sounds like microglia can function as calcium sensors but actually meant is that calcium signals from G-CaMP can be measured in microglia. or the abstract.